# Generation of mitochondrial reactive oxygen species is controlled by ATPase inhibitory factor 1 and regulates cognition

Pau B. Esparza-Moltó[1,2,3], Inés Romero-Carramiñana[1,2,3], Cristina Núñez de Arenas[1,2,3], Marta P. Pereira[1], Noelia Blanco[1,2,3], Beatriz Pardo[1], Georgina R. Bates[4,5], Carla Sánchez-Castillo[6], Rafael Artuch[2,7], Michael P. Murphy[4,5], José A. Esteban[6], José M. Cuezva[1,2,3]*

1 Departamento de Biología Molecular, Centro de Biología Molecular Severo Ochoa, Consejo Superior de Investigaciones Científicas-Universidad Autónoma de Madrid (CSIC-UAM), Madrid, Spain, 2 Centro de Investigación Biomédica en Red de Enfermedades Raras (CIBERER), ISCIII, Madrid, Spain, 3 Instituto de Investigación Hospital 12 de Octubre, Madrid, Spain, 4 MRC Mitochondrial Biology Unit, University of Cambridge, Cambridge, United Kingdom, 5 Department of Medicine, University of Cambridge, Addenbrooke's Hospital, Cambridge, United Kingdom, 6 Unidad de Neuropatología Molecular, Centro de Biología Molecular Severo Ochoa, Madrid, Spain, 7 Departamento de Bioquímica Clínica, Institut de Recerca Sant Joan de Déu, Barcelona, Spain

* jmcuezva@cbm.csic.es

**Data Availability Statement:** The authors confirm that all data underlying the findings are fully available without restriction. Gene expression

## Abstract

The mitochondrial ATP synthase emerges as key hub of cellular functions controlling the production of ATP, cellular signaling, and fate. It is regulated by the ATPase inhibitory factor 1 (IF1), which is highly abundant in neurons. Herein, we ablated or overexpressed IF1 in mouse neurons to show that IF1 dose defines the fraction of active/inactive enzyme in vivo, thereby controlling mitochondrial function and the production of mitochondrial reactive oxygen species (mtROS). Transcriptomic, proteomic, and metabolomic analyses indicate that IF1 dose regulates mitochondrial metabolism, synaptic function, and cognition. Ablation of IF1 impairs memory, whereas synaptic transmission and learning are enhanced by IF1 over-expression. Mechanistically, quenching the IF1-mediated increase in mtROS production in mice overexpressing IF1 reduces the increased synaptic transmission and obliterates the learning advantage afforded by the higher IF1 content. Overall, IF1 plays a key role in neuronal function by regulating the fraction of ATP synthase responsible for mitohormetic mtROS signaling.

## Introduction

The mitochondrial ATP synthase is the rotary engine of oxidative phosphorylation (OXPHOS) that utilizes the $H^+$ electrochemical gradient generated by the respiratory chain to synthesize most cellular ATP [1]. Moreover, the ATP synthase is also a component required for the efficient execution of cell death [2,3], and recent findings support that it significantly contributes to the permeability transition pore (PTP), which is the mitochondrial

microarray and proteomic data have been deposited to Gene Expression Omnibus (Project accession: GSE154064) and ProteomeXchange Consortium via the PRIDE partner repository (Project accession: PXD020262), respectively. Remaining relevant data are within the paper and its Supporting Information files.

**Funding:** P.B.E.-M and I.R.-C. were supported by predoctoral fellowships from Fundación La Caixa (Obra Social La Caixa, LCF/BQ/ES15/1036002) and FPU18/02234 from Ministerio de Educación Cultura y Deporte, Spain, respectively. The work was supported by grants from Ministerio de Economía y Competitividad (MINECO) (SAF2016-75916-R and PID2019-108674RB-I00) and CIBERER-ISCIII (CB06/07/0017), Spain, to J.M.C. The funders had no role in study design, data collection and analysis, decision to publish, or preparation of the manuscript.

**Competing interests:** I have read the journal's policy and the authors of this manuscript have the following competing interests: M.P.M. consults for Antipodean Pharmaceutical Inc., which is commercializing MitoQ. The rest of authors declare that they have no conflict of interest.

**Abbreviations:** ΔΨm, mitochondrial membrane potential; Δp, proton motive force; ACC, acetyl-CoA carboxylase; ACSF, artificial cerebrospinal fluid; AMPK, AMP-activated protein kinase; AP, anteroposterior; BN, blue native; BSA, bovine serum albumin; BSTFA, bis(trimethyl-silyl)trifluoro-acetamide; *Camk2a*, Calcium/calmodulin-dependent protein kinase II α; DIV, days in vitro; DRP1, dynamin-1-like protein; DV, dorsoventral; ECAR, extracellular acidification rate; EMMA, European Mouse Mutant Archive; ERK 1/2, extracellular signal-regulated kinase ½; FDR, false discovery rate; fEPSP, field excitatory postsynaptic potential; Fmr1, fragile X mental retardation 1; GABA$_A$, gamma-aminobutyric acid class A receptors; GCLC, glutamate cysteine ligase; GO, gene ontology; GSEA, gene set enrichment analysis; GSR, glutathione reductase; HMOX1, heme oxygenase 1; IF1, ATPase inhibitory factor 1; *IF1*$^{KO}$, IF1 knockout; *IF1*$^{TG}$, IF1 overexpressing transgenic; IMMT, mitofilin; IPA, ingenuity pathway analysis; iTRAQ, isobaric tags for relative and absolute quantitation; KEAP1, kelch-like ECH-associated protein 1; KEGG, Kyoto Encyclopedia of Genes and Genomes; LC–MS/MS, liquid chromatography with tandem mass spectrometry; LTP, long-term potentiation; ML, medio-lateral; MS/MS, tandem mass spectrometry; mtDNA, mitochondrial DNA; mtROS, mitochondrial reactive oxygen species; NRF2, nuclear factor erythroid 2-like 2; OCR, oxygen consumption rate; OXPHOS,

megachannel whose prolonged opening commits cells to death [4,5]. The mitochondrial ATP synthase also regulates life span extension as revealed in RNA interference (RNAi) screens for genes promoting longevity in worms [6,7] and flies [8] and when inhibited by mitochondrial oxoacids [9,10]. More recently, the ATP synthase has also emerged as a key hub in signaling mitohormetic programs through mitochondrial reactive oxygen species (mtROS) [11]. In fact, inhibition of the ATP synthase by the overexpression of the ATPase inhibitory factor 1 (IF1) confers increased resistance to toxic insults in different mouse tissues [12–14]. However, the expression of IF1 in mouse tissues that naturally do not express the protein is detrimental and results in metabolic syndrome [15] or in oncogenesis [14], strongly emphasizing the pivotal tissue-specific role of the mammalian ATP synthase/IF1 axis in cellular signaling.

IF1 is a structurally disordered protein that reversibly binds to the enzyme [16], blocking its rotatory catalysis [17]. For many years, IF1 has been considered to inhibit only the reverse activity of the enzyme to prevent the hydrolysis of cellular ATP when mitochondria become de-energized [18,19]. However, the overexpression of IF1 in cultured cells [20–24] or in mouse tissues [12–15] inhibits the ATP synthetic activity of the enzyme contributing to metabolic reprogramming to an enhanced glycolysis and the production of mtROS. Moreover, and in contrast to the idea that IF1 binds only to the ATP synthase upon acidification of the mitochondrial matrix [19,25], IF1 is found bound to tetramers of heart ATP synthase under physiological conditions [26,27], suggesting that IF1 regulates the activity of the enzyme under phosphorylating conditions. Certainly, phosphorylation of S39 in IF1 prevents its interaction with the ATP synthase, abolishing its inhibitory effect on the heart enzyme in vivo [28].

The contribution of IF1 to human diseases such as cancer and diabetes has been recently outlined [14,15,21,29], but its physiological relevance in the tissues where it is expressed is not well understood [30]. That is the case of human and mouse neurons, where IF1 levels exceed those of the ATP synthase [30]. Herein, we have generated mouse models expressing different levels of IF1 in neurons to study the functional relevance of the ATP synthase/IF1 in neuronal and higher-order brain functions. Overall, we provide a comprehensive genetic demonstration that a large fraction of neuronal IF1 binds to and inhibits the ATP synthase in vivo to regulate mitochondrial respiration and mtROS production. Moreover, the IF1-mediated mtROS signaling plays a key role in synaptic transmission and cognition. Overall, the findings highlight the ATP synthase/IF1 axis as a potential target to prevent cognitive impairments.

## Results

### Mouse models for the genetic regulation of the ATP synthase in neurons

To explore the biological role of IF1 in neurons, we have developed neuron-specific conditional IF1 knockout (*IF1*$^{KO}$) and IF1 overexpressing transgenic (*IF1*$^{TG}$) mice. IF1-floxed mice were generated from a mouse line harboring a reporter-tagged *Atp5if1* conditional allele (Atpif1$^{tm1a}$) after ubiquitous Flp-mediated deletion of the gene trapping cassette (Fig 1A and 1B). *IF1*$^{KO}$ mice were obtained by deleting exon 3 of IF1 in the floxed mice by Cre-mediated recombination under the control of Ca$^{2+}$/calmodulin-dependent protein kinase II α (*Camk2a*) promoter, which is active mostly in excitatory neurons (Fig 1A and 1B). *IF1*$^{TG}$ transgenic mice were developed by breeding mice harboring human IF1 under the control of a tetracycline-regulated element with mice expressing the transactivator under the control of *Camk2a* promoter (Fig 1B and 1C). IF1 is a highly conserved protein in mammals that inhibits the ATP synthase of different species, including yeast [31], and the overexpression of human IF1 is likely to recapitulate a gain of function of rodent IF1 [12,22]. IF1 expression was ablated in forebrain neurons of *IF1*$^{KO}$ mice (Fig 1D), and human IF1 was overexpressed in *IF1*$^{TG}$ mice, increasing total IF1 levels (Fig 1D), as revealed using an antibody that recognizes both the

oxidative phosphorylation; PCA, principal component analysis; PRDX3, peroxiredoxin 3; PRDX6, peroxiredoxin 6; PTP, permeability transition pore; qPCR, quantitative polymerase chain reaction; RET, reverse electron transfer; RhoA, Ras homolog family member A; RNAi, RNA interference; SOD, superoxide dismutase; TBS, Tris-buffered saline; tTa, tetracycline transactivator; UPLC, ultra-high-performance liquid chromatography; VDAC, voltage-dependent anion channel; WAVE1, WASP family member 1.

human and mouse proteins [30]. Moreover, IF1 knockout was specific for forebrain regions, not affecting the cerebellum or other tissues with high IF1 content (S1A Fig). Heterozygous mice with one functional copy of *Atp5if1* gene expressed normal levels of the protein (S1B Fig), suggesting the relevance of posttranscriptional mechanisms for the regulation of IF1 expression [30].

Importantly, the ATP synthetic (Fig 1E) and hydrolytic (Fig 1F) activities in isolated mitochondria increased upon ablation of IF1 and were significantly reduced as the dose of IF1 increased, suggesting that IF1 is bound to and inhibits a significant fraction of the ATP synthase under normal physiological conditions. To verify this point, ATP synthase was immunoprecipitated and the co-immunoprecipitated IF1 protein was determined (Fig 1G). The results revealed that the reduction in both enzyme activities correlated with higher levels of IF1 co-immunoprecipitated with the enzyme (Fig 1G), strongly supporting a role for IF1 in the regulation of the ATP synthase in neurons.

## IF1 dose regulates mitochondrial structure, respiration, and mtROS production

Electron microscopy analysis of the CA1 region of the hippocampus revealed that mitochondria of the neuronal soma from $IF1^{KO}$ mice were more rounded and showed less organized cristae architecture when compared to the more elongated mitochondria of control and $IF1^{TG}$ mice, although no differences were found in mitochondrial area (Fig 1H). At the molecular level, these changes were paralleled by a significant down-regulation of the pro-fission dynamin-1-like protein (DRP1) only in $IF1^{TG}$ mice (Fig 1I). Concurrently, the levels of voltage-dependent anion channel (VDAC) and of the core MICOS subunit mitofilin (IMMT) were increased in $IF1^{TG}$ mice (Fig 1I), suggesting that IF1 dose affects mitochondrial structure in neurons. However, no significant differences were found in mitochondrial DNA (mtDNA) copy number among the 3 genotypes (Fig 1J).

Both basal and oligomycin-sensitive respiration in primary cultures of hippocampal neurons were significantly reduced as the dose of IF1 augmented (Fig 2A and 2B), consistent with the IF1-mediated inhibition of the ATP synthase. Of note, maximal respiration was significantly reduced only in $IF1^{TG}$ mice (Fig 2A and 2B), due to a decrease in the activity of complex IV (Fig 2C) but not of complex I (Fig 2D), in agreement with previous findings [12]. These changes occurred in the absence of significant differences in markers of mitochondrial mass, such as mtDNA copy number (Fig 1J) and HSP60 content (Fig 1I), or in the expression (S1C Fig) and assembly (S1D Fig) of respiratory complexes, despite the large differences in IF1 expression (S1C Fig). The reduction of complex IV activity was not accompanied by the alteration of its supramolecular organization (S1D Fig), as shown in a previous report [12], because the wild-type IF1 exerts less stringent inhibition of the ATP synthase than the constitutively active IF1 mutant [22] that was used in that study [12]. However, IF1 dose significantly increased the oligomeric assemblies of ATP synthase (S1D Fig), in agreement with previous observations [14,19]. Interestingly, the H⁺ leak-linked respiration was significantly reduced with higher IF1 dose (Fig 2A and 2B), suggesting more efficient OXPHOS.

Both the lactate production (Fig 2E) and extracellular acidification rate (ECAR) (Fig 2F) rates in the primary cultures were reduced in $IF1^{TG}$ mice, suggesting a reduced glycolytic flux. Moreover, the maximal glycolytic capacity was also reduced in $IF1^{TG}$ mice, as assessed by ECAR upon oligomycin addition (Fig 2F). No significant differences were noted in the expression of glycolytic enzymes in brain extracts (Fig 2G). Although the phosphorylation of the AMP-activated protein kinase (AMPK) and of its downstream target acetyl-CoA carboxylase (ACC) was increased in the hippocampus of $IF1^{TG}$ mice (Fig 2G), the forebrain ATP/ADP

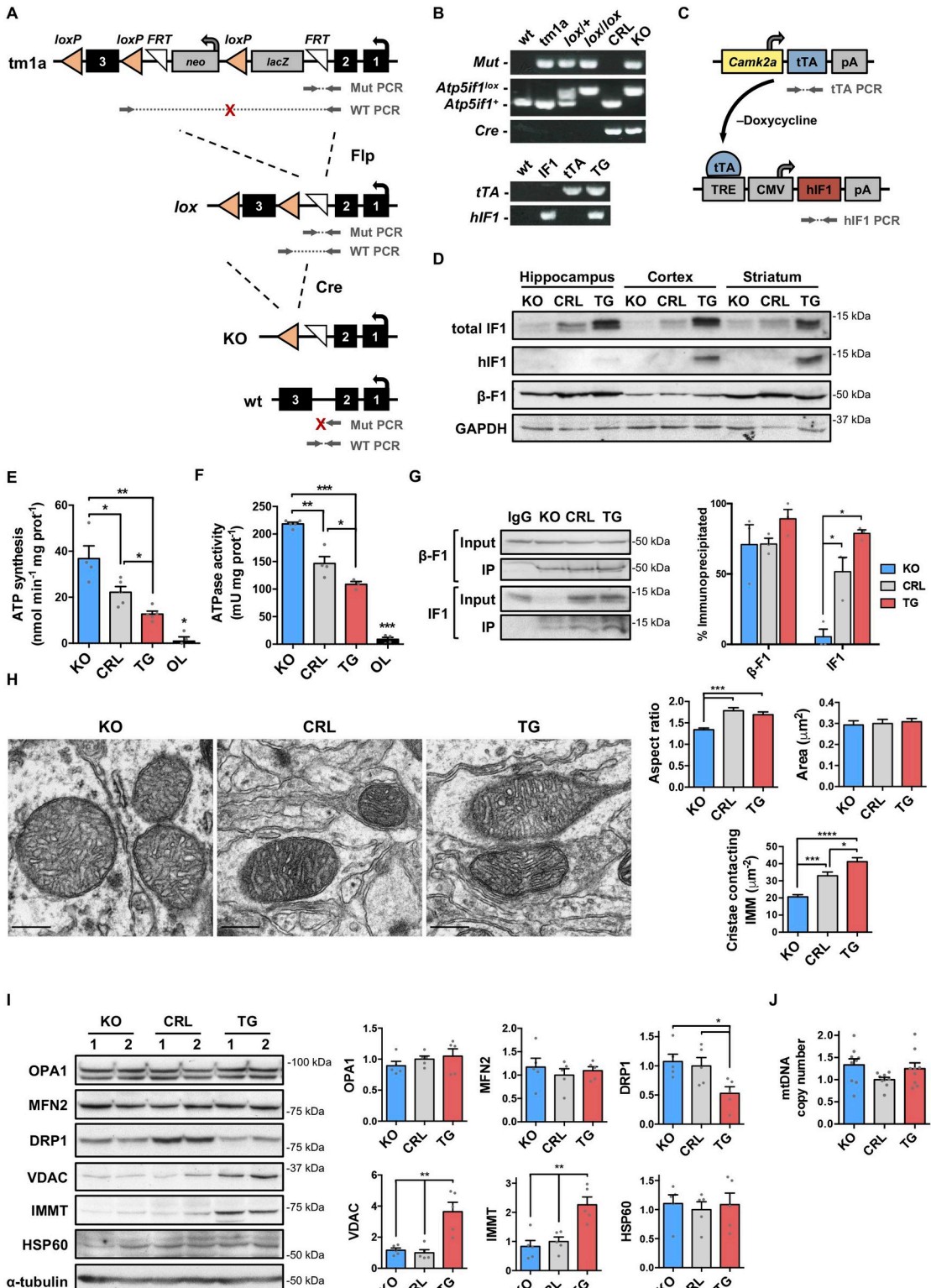

**Fig 1. Development of mouse models for the genetic regulation of the ATP synthase in neurons.** (**A**) Generation of $IF1^{KO}$ mice in neurons showing the *Atp5if1* knockout-first (tm1a), floxed (*lox*), KO, and wt alleles. The regions amplified by the PCRs used for genotyping are indicated. Arrows show primers, and red crosses indicate nonproductive PCRs. (**B**) PCR genotyping of wt, *lox*, CRL, and $IF1^{KO}$ mice (upper panel), or wt, IF1, tTA, and $IF1^{TG}$ mice (lower panel). (**C**) Schematic of *Camk2a-tTA* and *TRE-hIF1* constructs used to drive hIF1 expression in forebrain neurons in a tetracycline-inducible manner. (**D**) Western blot of total

(human and mouse) and hIF1 in extracts from different forebrain regions of $IF1^{KO}$, CRL, and $IF1^{TG}$ mice. GAPDH is shown as CRL. **(E and F)** ATP synthetic and hydrolytic activities in isolated forebrain mitochondria of $IF1^{KO}$, CRL, and $IF1^{TG}$ mice ($n = 3$–$5$) using succinate as a respiratory substrate. OL reduced both activities to a similar extent in the 3 genotypes (the average is shown). **(G)** Mitochondria from $IF1^{KO}$, CRL, and $IF^{TG}$ mice were IP with anti-$F_1$-ATPase, and the co-IP IF1 was identified by western blotting. The histogram to the bottom shows the quantification of IP β-F1 and IF1 ($n = 3$). **(H)** Representative electron micrographs of mitochondria in the CA1 region of the hippocampus (scale bar, 300 nm). Histograms to the right show the quantification of mitochondrial aspect ratio ($n = 46$–$69$ mitochondria each) and the number of cristae contacting the IMM normalized per mitochondrial area ($n = 18$–$22$) in 2 mice per genotype. **(I)** Western blots of OPA1, MFN2, DRP1, VDAC, IMMT, and HSP60 in hippocampal extracts. Histograms to the right show the quantification as fold change of CRL ($n = 5$). **(J)** mtDNA copy number in DNA extracted from forebrain of $IF1^{KO}$, CRL, and $IF1^{TG}$ mice ($n = 9$). Error bars: mean ± SEM. $^*P < 0.05$, $^{**}P < 0.01$, $^{***}P < 0.001$, $^{****}P < 0.0001$ by 2-tailed $t$ test (E–G and J) or Kruskal–Wallis with Dunn multiple comparisons test (I). Uncropped western blots can be found in S1 Raw Images, and numerical data underlying plots in S1 Data. CMV, minimal cytomegalovirus promoter; CRL, control; DRP1, dynamin-1-like protein; hIF1, human IF1; HSP60, 60-kDa heat-shock protein; IF1, ATPase inhibitory factor 1; $IF1^{KO}$, IF1 knockout; $IF1^{TG}$, IF1 overexpressing transgenic; IMM, inner mitochondrial membrane; IMMT, mitofilin; IP, immunoprecipitated; KO, knockout; MFN2, mitofusin 2; mtDNA, mitochondrial DNA; OL, oligomycin; OPA, optic atrophy 1; pA, polyadenylation signal; TRE, tetracycline response element; tTA, tetracycline transactivator; VDAC, voltage-dependent anion channel; wt, wild-type.

ratio was similar among the 3 genotypes (Fig 2H). Overall, these results suggest a reduced metabolic activity in neurons from $IF1^{TG}$ mice when compared to the other genotypes.

Importantly, the mitochondrial membrane potential (ΔΨm) and the production of mtROS increased in neuronal cultures as the dose of IF1 augmented, similarly to the effect of the ATP synthase inhibitor oligomycin (Fig 2I). Mitochondrial hyperpolarization and enhanced mtROS production are consistent with a reduced consumption of the proton motive force (Δp) due to inhibition of ATP synthesis by IF1 [20,22]. We investigated mtROS production by reverse electron transfer (RET) [32,33] through complex I in isolated brain mitochondria using succinate to reduce the CoQ pool and generate Δp [34]. Interestingly, mitochondria of the 3 genotypes showed a substantial rate of $H_2O_2$ production that was sensitive to—and similarly affected by—the complex I inhibitor rotenone (S1E Fig). These findings indicate that basal RET is the same among the 3 genotypes and emphasize that a relevant proportion of mtROS are produced by RET in brain mitochondria. Anyway, since RET levels in vivo are controlled by ΔΨm [34], the findings support that RET enhances its contribution to the mtROS production observed in neurons with higher IF1 dose (Fig 2I). Consistent with these observations, the antioxidant enzymes catalase, superoxide dismutase 1 and 2 (SOD1 and SOD2) were up-regulated in the hippocampus of $IF1^{TG}$ mice, supporting in vivo an increase in ROS production (Fig 2J). Of note, this increase should be mild, because the expression of other proteins of the antioxidant defense was not induced (Fig 2J). Overall, these findings support that IF1 dose plays an important role in the regulation of mitochondrial respiration and in the generation of mtROS in hippocampal neurons.

## IF1 dose regulates transcriptional programs of neuronal function

Changes in mtROS production regulate kinases and transcription factors involved in mitohormetic responses that favor adaptation to changing cues [35]. In order to unveil potential mechanisms regulated by IF1 dose, we carried out a transcriptomic analysis of the hippocampus and identified 463 differentially expressed genes between $IF1^{TG}$ and $IF1^{KO}$ mice (Fig 3A–3C). We focused on the $IF1^{TG}$ versus $IF1^{KO}$ comparison because it yielded the greatest number of differentially expressed genes, which included genes from the other 2 comparisons (Fig 3A and Data Availability GSE154064). Principal component analysis (PCA) revealed the clustering of $IF1^{TG}$ mice away from $IF1^{KO}$ and control counterparts (Fig 3D). As shown by gene ontology (GO) analysis, most differentially expressed genes encoded proteins localized outside mitochondria (Fig 3E). Moreover, gene set enrichment analysis (GSEA) confirmed the upgrading of genes related to cellular communication and postsynaptic membrane in $IF1^{TG}$

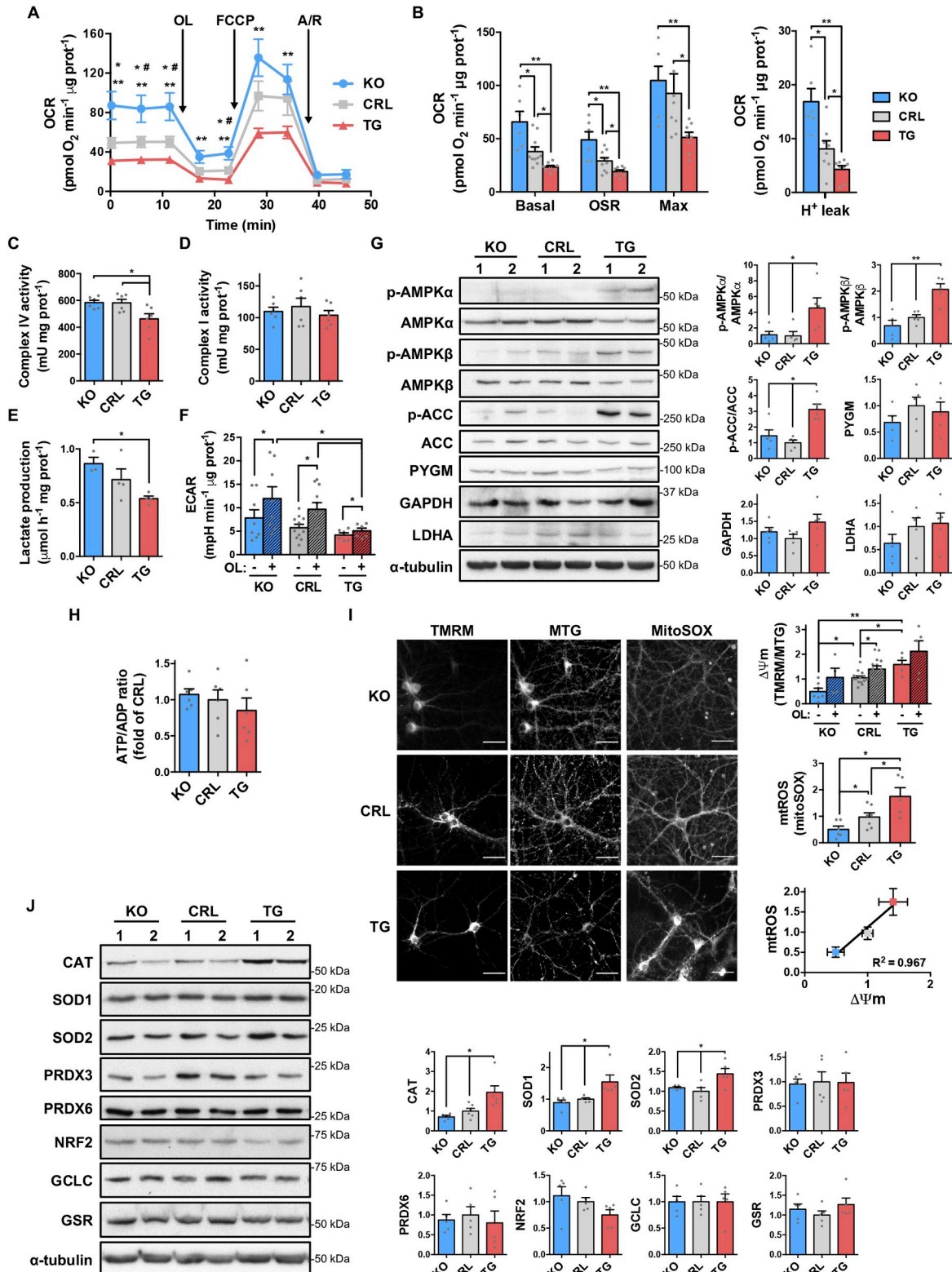

**Fig 2. IF1 regulates mitochondrial respiration and ROS production in neurons.** (**A**) Respiratory profiles of $IF1^{KO}$ (blue line), CRL (gray), and $IF1^{TG}$ (red) primary neurons. The addition of OL, FCCP, and A/R is indicated. (**B**) The histograms show basal, OSR, maximal and proton leak-linked respiration rates in Seahorse XF24 ($n$ = 6–8 embryos from 3 independent cultures). (**C and D**) Complexes IV and I activities in isolated forebrain mitochondria ($n$ = 6). (**E and F**) Glycolytic activity measured by the rate of lactate production or ECAR in

primary cultures ($n$ = 6–11 embryos from 3 independent cultures). **(G)** Western blots of the phosphorylation and expression of AMPKα and AMPKβ, ACC, PYGM, GAPDH and LDHA in hippocampal extracts. α-tubulin is shown as control, and 2 representative samples are shown (1 and 2). Histograms to the right show the quantification as fold change of CRL ($n$ = 5). **(H)** ATP/ADP ratio in the forebrain ($n$ = 6). **(I)** Staining of primary neurons with TMRM, MTG, and MitoSOX (scale bar, 40 μm). Histograms to the right show the quantification of ΔΨm and mtROS as fold change of CRL ($n$ = 4–14 embryos) and their positive correlation. **(J)** Western blots of CAT, SOD1 and SOD2, PRDX3 and PRDX6, NRF2, the catalytic subunit of the GCLC, and GSR in hippocampal extracts. α-tubulin is shown as control. Histograms to the right show the quantification as fold change of CRL ($n$ = 5). Error bars: mean ± SEM. $^*P < 0.05$, $^{**}P < 0.01$ by 2-tailed $t$ test (B–J). In (A), $^*P < 0.05$ for $IF1^{KO}$-CRL, $^\#P < 0.05$ for CRL-$IF1^{TG}$, and $^{**}P < 0.01$ for $IF1^{KO}$–$IF1^{TG}$ comparisons (2-tailed $t$ test). See also S1 Fig. Uncropped western blots can be found in S1 Raw Images, and numerical data underlying plots in S1 Data. ΔΨm, mitochondrial membrane potential; ACC, acetyl-CoA carboxylase; AMPKα, AMP-activated protein kinase subunit alpha; AMPKβ, AMP-activated protein kinase subunit beta; A/R, antimycin A and rotenone; CAT, catalase; CRL, control; ECAR, extracellular acidification rate; GSR, glutathione reductase; IF1, ATPase inhibitory factor 1; $IF1^{KO}$, IF1 knockout; $IF1^{TG}$, IF1 overexpressing transgenic; KO, knockout; LDHA, lactate dehydrogenase A; MTG, MitoTracker Green; mtROS, mitochondrial reactive oxygen species; NRF2, nuclear factor erythroid 2-like 2; OL, oligomycin; OSR, oligomycin-sensitive respiration; PRDX, peroxiredoxin; PYGM, glycogen phosphorylase; ROS, reactive oxygen species; SOD, superoxide dismutase.

mice (Fig 3F). These findings support that IF1 dose, besides regulating mitochondrial structure and activity (Figs 1 and 2), also affects other neuronal structures such as dendrites and post-synaptic membranes, which are crucial for synaptic transmission. In agreement with this idea, ingenuity pathway analysis (IPA) predicted differences in dendrite dynamics, synaptic plasticity, short-term memory, and behavior (Fig 3G). Moreover, signaling by the tumor necrosis factor receptor family member CD40, which promotes neuroinflammation [36], was reduced in $IF1^{TG}$ mice (Fig 3G). Quantitative polymerase chain reaction (qPCR) analysis of key genes found differentially expressed among the 3 genotypes confirmed the transcriptomic study (Fig 3H).

## Proteomic and metabolomic analyses confirm the role of IF1 in neuronal function

A quantitative proteomic analysis of the hippocampus of $IF1^{KO}$, control, and $IF1^{TG}$ mice identified 6,168 peptides, corresponding to 58 different proteins, which were differentially expressed among the 3 genotypes (Fig 4A) and clearly separated them (Fig 4B). The expression of some proteins correlated with IF1 dose (Fig 4C and 4D). As observed with the transcriptomic data, differentially expressed proteins were located both in mitochondria and in other compartments including neuron projections (Fig 4E) and showed significant enrichment for learning or memory in $IF1^{TG}$ mice (Fig 4F). We found a subtle down-regulation mice of OXPHOS proteins in $IF1^{TG}$ (Fig 4G), which included COX6B1 subunit of mitochondrial complex IV, that could contribute to the reduced cytochrome $c$ oxidase activity observed in these mice (Fig 2C), of Rieske protein (UQCRFS1) and of $\delta$, $b$, and $f$ subunits of the ATP synthase (see Data Availability PXD020262). Long-term potentiation (LTP), a mechanism of synaptic plasticity underlying learning and memory [37], was also found enriched among the down-regulated proteins in $IF1^{TG}$ mice (Fig 4G). On the other hand, up-regulated proteins in $IF1^{TG}$ mice showed enrichment for enzymes involved in the metabolism of alanine, aspartate, glutamate, and glutathione (Fig 4G), namely glutamine synthetase, glutamate decarboxylase, and 4-aminobutyrate aminotransferase (see Data Availability PXD020262).

Interestingly, a metabolomic analysis of the hippocampus (Fig 5A, S1 Table) revealed that glutamate was significantly reduced in $IF1^{TG}$ mice (Fig 5B), consistent with the up-regulation of the enzymes that consume glutamate (Fig 4G). Glutamate is the main excitatory neurotransmitter in the brain, and its reduction in $IF1^{TG}$ mice (Fig 5B) reinforces the existence of differences in synaptic function as suggested by the omic analyses (Figs 3G and 4G). Glutamine, aspartate, and alanine were also reduced in $IF1^{TG}$ mice (Fig 5B), what could suggest that the

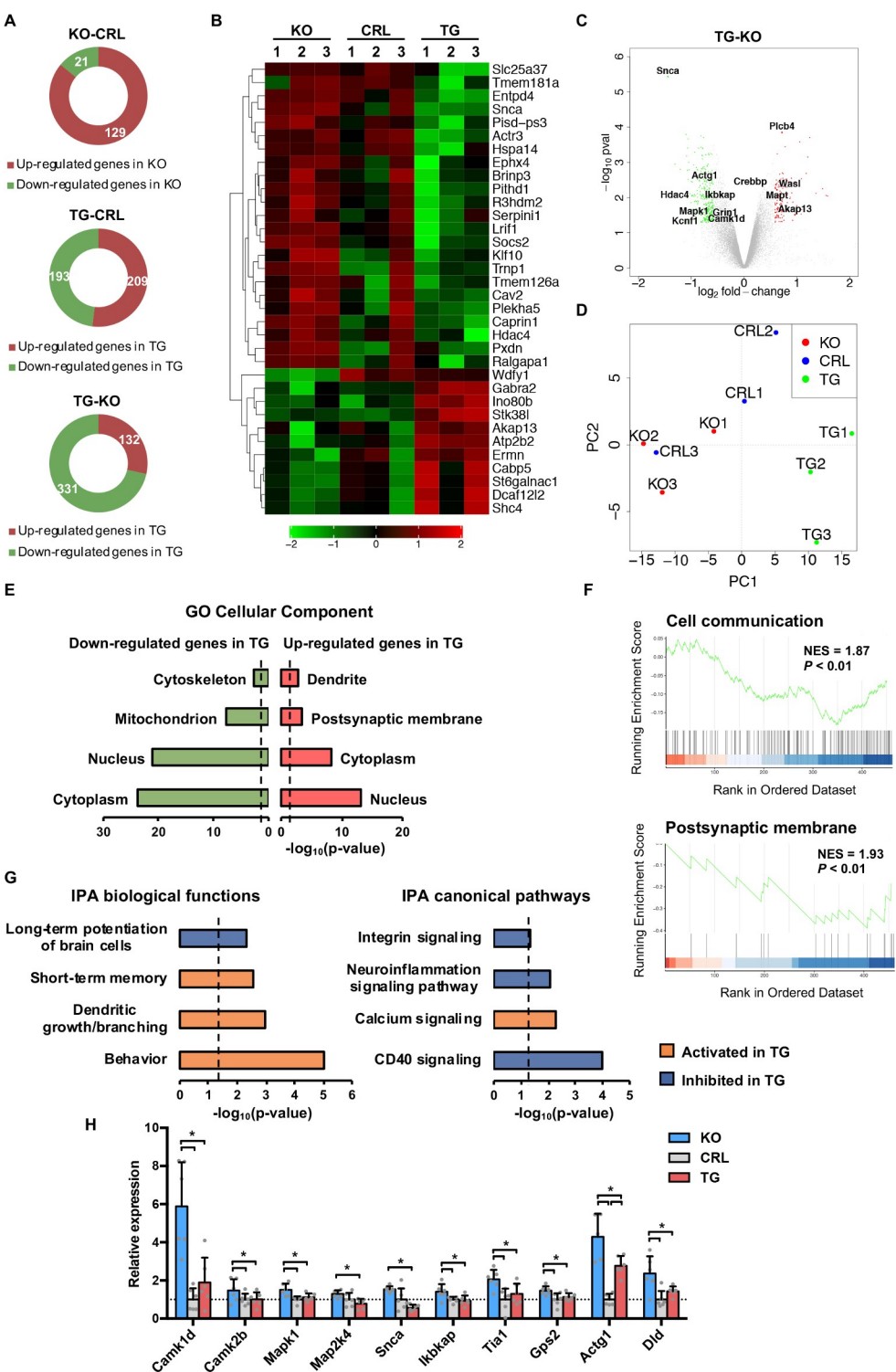

**Fig 3. IF1 affects gene expression programs of neuronal function.** (A) Differentially expressed genes identified in the 3 comparisons among *IF1^KO*, CRL, and *IF1^TG* mice. (B) Heatmap showing genes exhibiting statistically significant (*P* < 0.05) and 2-fold or greater increases between *IF1^KO* and *IF1^TG* mice. (C) Volcano plot showing differentially expressed genes in *IF1^TG* vs. *IF1^KO* comparison. (D) PCA of the 3 genotypes. (E) Gene enrichment analysis of the down- and up-regulated transcripts in *IF1^TG* with respect to *IF1^KO* mice, showing the GO cellular components. The dashed line shows the significance threshold. (F) Enrichment plots of top hits obtained from GSEA. (G) IPA of biological functions and canonical pathways with their predicted activation/inhibition status. The dashed lines show

the significance threshold. **(H)** Quantification by real-time PCR of key differentially expressed genes identified in the array. Error bars: mean ± SEM of 6 mice per genotype. *$P < 0.05$ by 2-tailed $t$ test. Numerical data underlying plots can be found in S1 Data. CRL, control; GO, gene ontology; GSEA, gene set enrichment analysis; IF1, ATPase inhibitory factor 1; $IF1^{KO}$, IF1 knockout; $IF1^{TG}$, IF1 overexpressing transgenic; IPA, ingenuity pathway analysis; KO, knockout; PCA, principal component analysis.

up-regulated proteins (Fig 4G) respond to a compensatory deficit of the metabolites (Fig 5B). We observed no relevant changes in glutathione levels with increasing IF1 dose (Fig 5B), perhaps because the mild antioxidant response triggered by the up-regulation of catalase, SOD1, and SOD2 (Fig 2J) is sufficient to buffer mtROS production. The metabolism of aromatic amino acids was also enriched in the dataset (Fig 5A and 5C). Specifically, phenylalanine and tyrosine, which are precursors of the neurotransmitter dopamine, dopamine itself, and its metabolites, were significantly reduced in $IF1^{TG}$ mice (Fig 5C). Of note, the reduction in the tissue content of a large number of amino acids in $IF1^{TG}$ mice (Fig 5, S1 Table) suggests that IF1 contributes to the reprogramming of amino acid metabolism in neurons.

## IF1 overexpression increases basal synaptic transmission

The combined structural, functional, and omic analyses suggested that changes in mitochondrial IF1 expression might affect synaptic activity. To test this possibility, we determined whether IF1 dose affects basal transmission in the glutamatergic synapse between CA3 and CA1 pyramidal neurons, which is key for learning and memory [37]. Electrophysiological recordings were performed in acute hippocampal slices using a stimulating electrode placed over Schaffer collateral fibers in the stratum radiatum of the CA1 region (Fig 6A) [38]. The slope of field excitatory postsynaptic potentials (fEPSPs) (Fig 6B) was measured to monitor synaptic activation of the CA1 pyramidal neurons. Interestingly, a 3-fold higher postsynaptic response was observed over a range of stimulatory intensities in $IF1^{TG}$ mice when compared to the other genotypes (Fig 6C). Notably, this was also reflected by the presence of population spikes, which appeared at high stimulation intensities in responses from $IF1^{TG}$ slices, and are caused by the firing of action potentials in the postsynaptic neurons due to their higher stimulation (see representative trace in Fig 6C).

The increased basal synaptic transmission in $IF1^{TG}$ mice correlated with greater phosphorylation of the GluA1 subunit of AMPA receptors (Fig 6D), which is associated with a higher opening probability [39] and channel conductance [40]. The phosphorylation of extracellular signal-regulated kinase 1/2 (ERK 1/2) (Fig 6D), whose signaling increases synaptic transmission [41], was also increased. Interestingly, although spine density was lower in $IF1^{TG}$ neurons (Fig 6E), spines were significantly larger when compared to the other genotypes (Fig 6E), which could be related to the increased basal transmission in these animals (Fig 6C) [42]. Differences in spine dynamics could be partially explained by differential organization of the actin cytoskeleton [43], as suggested by the transcriptomic and proteomic analyses (Figs 3E, 3G and 4G). Indeed, the expression of the GTPase Ras homolog family member A (RhoA), which promotes spine growth [44], was increased in the hippocampus of $IF1^{TG}$ mice (Fig 6F). In contrast, WASP family member 1 (WAVE1), whose activity increases spine density [45], was down-regulated (Fig 6F). Strikingly, there were no significant differences in LTP expression among the 3 genotypes (S2A Fig), despite a strong decrease in the levels of NMDA receptor subunits in $IF1^{TG}$ mice (S2B Fig), which mediate the main form of LTP at CA1 synapses [46]. No significant down-regulation of the mRNA levels of NMDA receptor subunits was found in $IF1^{TG}$ mice (S2C Fig and Data Availability GSE154064). Hence, the decrease in their

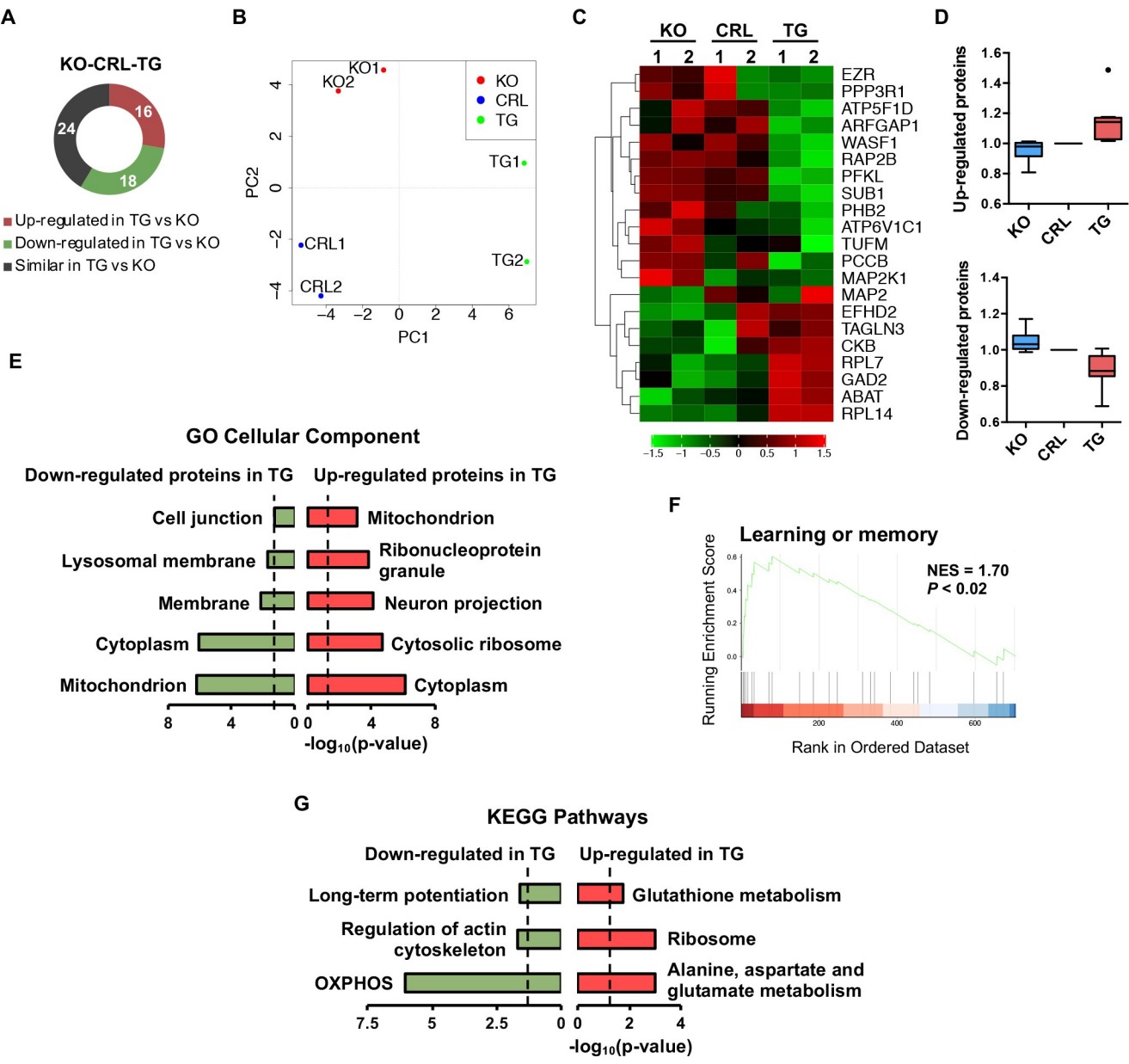

**Fig 4. IF1 dose affects the proteome of hippocampal neurons. (A)** Differentially expressed proteins among *IF1*<sup>KO</sup>, CRL, and *IF1*<sup>TG</sup> mice resulting from the iTRAQ analysis. **(B)** PCA of the 3 genotypes. **(C)** Heatmap depicting the proteins whose expression correlates with IF1 dose. **(D)** Box plots showing the expression of the proteins that increase or decrease with higher IF1 dose, expressed as fold of CRL (*n* = 8 up-regulated and *n* = 13 down-regulated proteins; center line, median; box limits, upper and lower quartiles; whiskers, 1.5× interquartile range; points, outlier). **(E)** Gene enrichment analysis of the down- and up-regulated proteins in *IF1*<sup>TG</sup> with respect to *IF1*<sup>KO</sup> mice, showing the significant GO cellular c. **(F)** Enrichment plot of a top hit obtained in GSEA from proteomic data. **(G)** Gene enrichment analysis of the down- and up-regulated proteins in *IF1*<sup>TG</sup> vs. *IF1*<sup>KO</sup> mice, showing the KEGG pathways significantly enriched. Numerical data underlying plots can be found in S1 Data. CRL, control; GSEA, gene set enrichment analysis; IF1, ATPase inhibitory factor 1; *IF1*<sup>KO</sup>, IF1 knockout; *IF1*<sup>TG</sup>, IF1 overexpressing transgenic; KEGG, Kyoto Encyclopedia of Genes and Genomes; KO, knockout; OXPHOS, oxidative phosphorylation; PCA, principal component analysis.

expression in *IF1*<sup>TG</sup> mice could result from their internalization and ubiquitin-dependent degradation [47], as previously reported for GluN1 subunit upon prolonged increases in neuronal activity in vitro by the treatment with bicuculline, an antagonist of the inhibitory GABA$_A$ receptors [48].

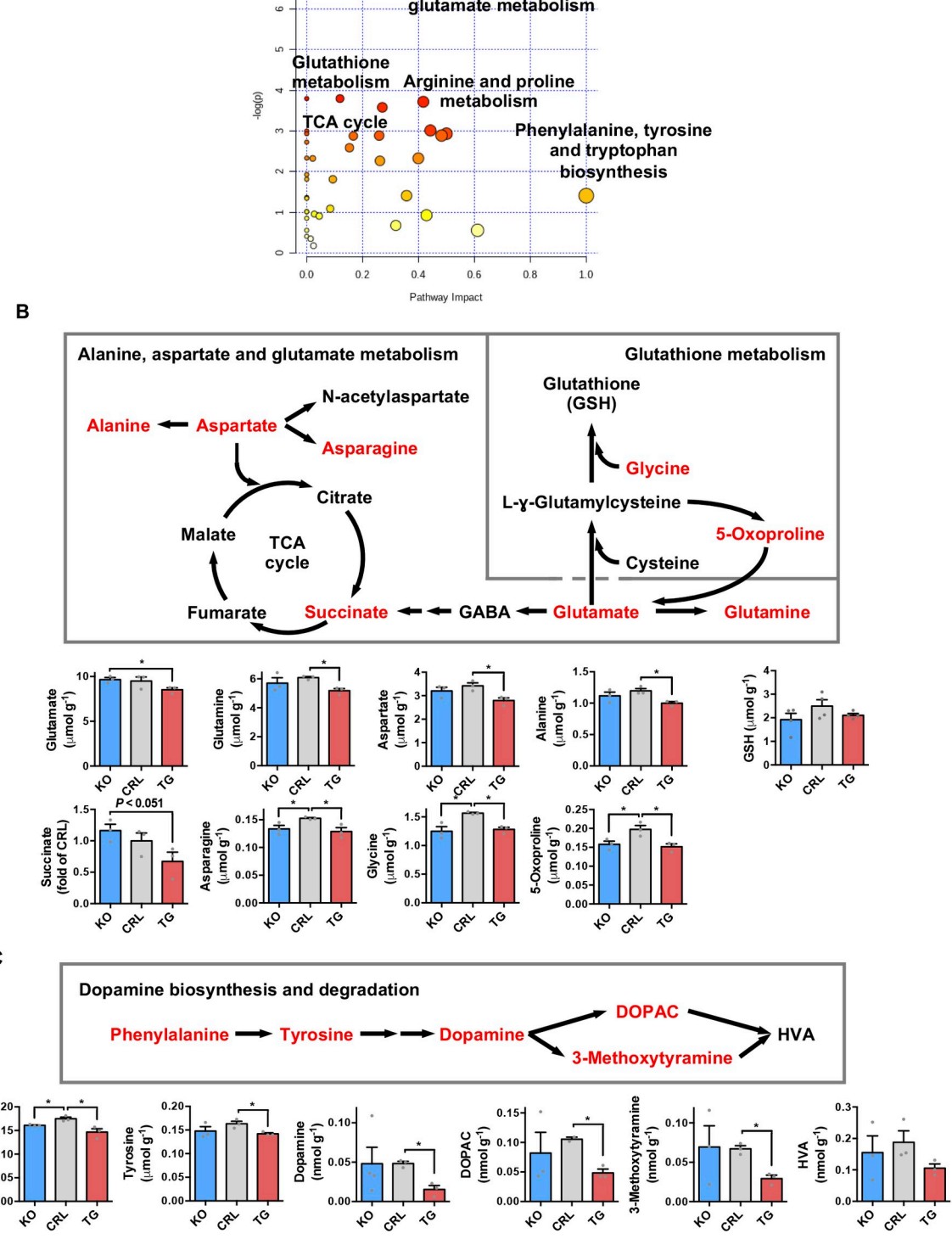

**Fig 5. IF1 dose alters the metabolism of glutamate and dopamine in the hippocampus.** (**A**) Pathway enrichment analysis of the metabolites in the comparison *IF1^TG^* vs. *IF1^KO^* mice. The node color is based on the *P* values from integrated enrichment analysis and the node radius represents the pathway impact values from topology analysis. (**B**) Overview of alanine, aspartate and glutamate, and GSH metabolic pathways. Metabolites significantly reduced in *IF1^TG^* vs. CRL mice are shown in red, and their quantification is shown to the bottom (*n* = 3). (**C**) Schematic of the biosynthesis and degradation of dopamine. Metabolites significantly reduced in *IF1^TG^* vs. CRL mice

are shown in red, and their quantification is shown to the bottom ($n$ = 3). Error bars: mean ± SEM. $^*P < 0.05$ by 2-tailed $t$ test. See also S1 Table. Numerical data underlying plots can be found in S1 Data. CRL, control; DOPAC, 3,4-dihydroxyphenylacetic acid; GABA, γ-aminobutyric acid; GSH, glutathione; HVA, homovanillic acid; IF1, ATPase inhibitory factor 1; $IF1^{KO}$, IF1 knockout; $IF1^{TG}$, IF1 overexpressing transgenic; KO, knockout; TCA cycle, tricarboxylic acid cycle.

## IF1 dose affects learning

Given the omic (Figs 3G and 4G) and synaptic differences among the 3 genotypes (Fig 6C), we next examined whether the differential expression of IF1 may have broader effects on behavior and cognition. In the open field test, $IF1^{KO}$ mice showed reduced basal exploratory activity when compared to the other 2 genotypes (Fig 6G and 6H). Memory was assessed in the novel object recognition test (Fig 6I), which relies on the rodents' tendency to preferentially explore novel objects [49]. Of note, all mice participated equally in these tests (S2C Fig). In the short-term memory test, control and $IF1^{TG}$ mice showed discrimination indexes higher than 50%, indicating that they recognized the novel object (Fig 6J). Notably, $IF1^{KO}$ mice exhibited impaired cognition with failure to recognize novelty (Fig 6J) and tended to explore both objects more often when compared to CRL and $IF1^{TG}$ (S2C Fig). Interestingly, in the long-term memory test, only $IF1^{TG}$ mice recognized the novel object (Fig 6K). These findings suggest that $IF1^{KO}$ mice have impaired short-term memory, while the expression of IF1 in neurons enhances memory as a function of IF1 dose.

The enhanced long-term memory of $IF1^{TG}$ mice was partially resistant against a scopolamine-based amnesic protocol interfering with memory consolidation [50] (S2D–S2F Fig). These results reinforce the better cognitive performance of $IF1^{TG}$ mice and together support a functional role for IF1 in neurons, because its ablation impairs learning while learning is enhanced by its overexpression.

## mtROS signaling mediates the increased synaptic transmission and memory of $IF1^{TG}$ mice

ROS regulate neuronal development and function, including synaptic activity [51] and the activation of ERK 1/2 [52]. Next, we investigated whether mtROS contribute to enhanced cognition in $IF1^{TG}$ mice. Administration of the mitochondria-targeted antioxidant MitoQ to $IF1^{TG}$ mice reduced the hippocampal level of mtROS in living mice, as measured using the mitochondria-targeted exomarker MitoB, whereby an increase in mtROS leads to an increase in the ratio of MitoP relative to MitoB (Fig 7A), suggesting the partial quenching of mtROS production in vivo by MitoQ [53]. This point was further confirmed by the decreased expression of some antioxidant defense proteins upon MitoQ treatment, which included nuclear factor erythroid 2-like 2 (NRF2), its regulator kelch-like ECH-associated protein 1 (KEAP1), and some of the downstream targets such as the mitochondrial enzymes SOD2 and peroxiredoxin 3 (PRDX3) (Fig 7B). Interestingly, MitoQ did not affect the expression of antioxidant proteins located outside mitochondria, such as catalase, SOD1, peroxiredoxin 6 (PRDX6), glutathione reductase (GSR), and heme oxygenase 1 (HMOX1) (Fig 7B). MitoQ administration also reduced protein carbonylation and tyrosine 3-nitration in hippocampal extracts (S3A and S3B Fig).

Electrophysiological recordings in hippocampal slices revealed that MitoQ partially reduced the enhanced basal synaptic transmission found in $IF1^{TG}$ mice (Fig 7C). The phosphorylation of ERK 1/2 and of Camk2a was also reduced upon MitoQ treatment (Fig 7D), supporting a role for mtROS production in activating the signaling pathways controlling synaptic function. However, we did not observe changes in the phosphorylation of AMPA receptor subunit

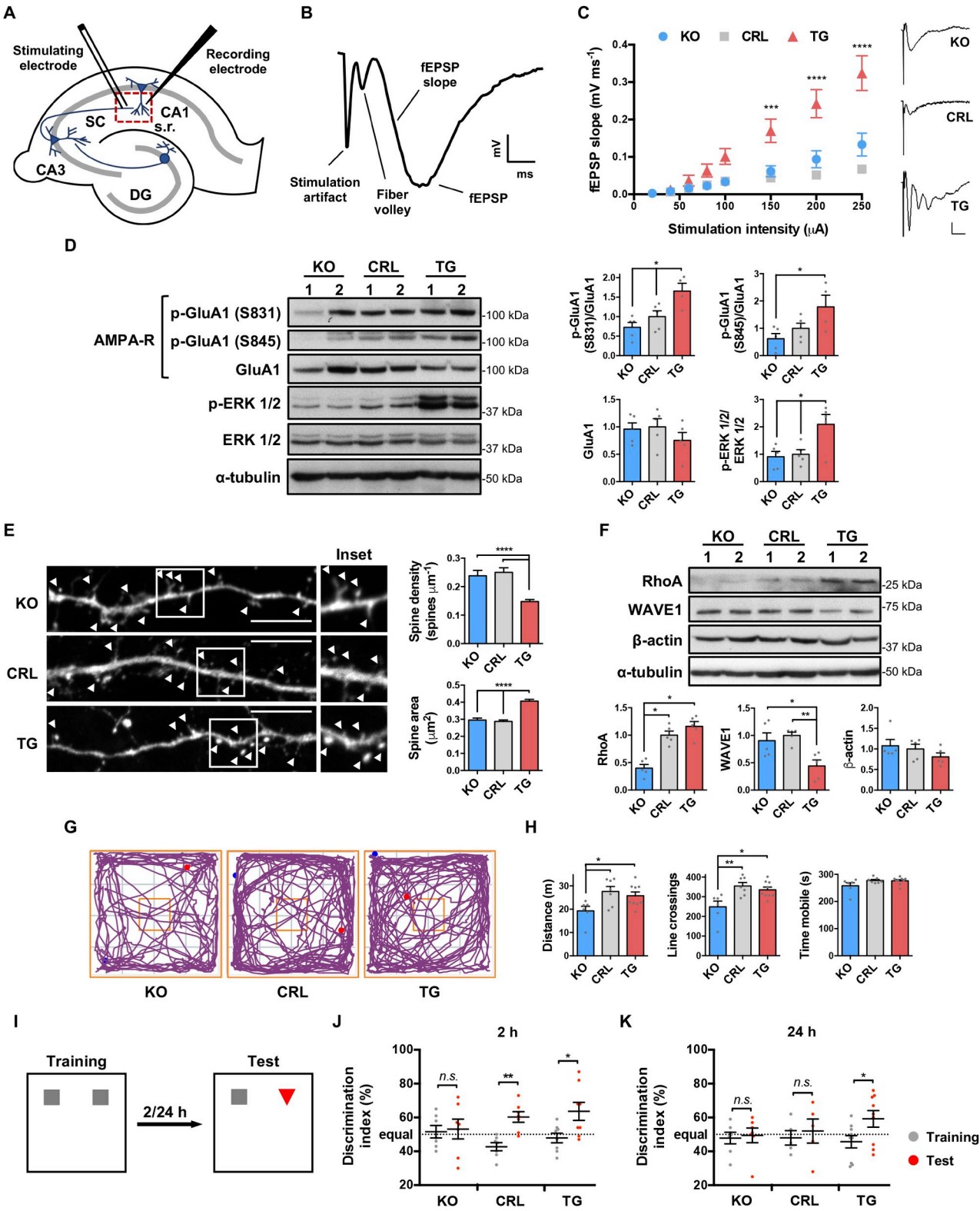

**Fig 6. IF1 plays a fundamental role in synaptic and cognitive functions. (A)** Configuration of the field recordings performed in hippocampal slices. **(B)** Representative trace to illustrate an extracellular recording in the stratum radiatum of the CA1 region (no scale). **(C)** Input–output curves showing fEPSP slope in slices from $IF1^{KO}$ ($n = 8$), CRL ($n = 9$), and $IF1^{TG}$ ($n = 11$) mice ($n = 5$ mice each) in response to increasing stimulus strength. Insets: representative fEPSPs at 250 μA (scale bar of the insets, 0.2 mV, 10 ms). **(D)** Western blots of the phosphorylation and expression of GluA1 and ERK

1/2. α-tubulin is shown as control, and 2 representative samples are shown (1 and 2). Histograms to the right show the quantification as fold of control CRL mice ($n = 5$). **(E)** Representative images of dendritic spines (arrowheads) from primary hippocampal neurons transfected with EGFP (scale bar, 10 μm). Histograms to the right show the quantification of spine density and area ($n = 247$–$628$ spines from 2 independent cultures). **(F)** Western blots of RhoA, WAVE1, and β-actin in hippocampal extracts. Histograms to the bottom show the quantification as fold of control CRL mice ($n = 5$). **(G)** Representative track plots of $IF1^{KO}$, CRL, and $IF1^{TG}$ mice in the open field test. Blue and red dots indicate the start and end point of the track, respectively. **(H)** Exploratory activity of $IF1^{KO}$ ($n = 6$), CRL ($n = 7$), and $IF1^{TG}$ mice ($n = 9$). **(I)** Schematic of the novel object recognition test. **(J and K)** Plots show the discrimination index at 2 and 24 hours after the training session. $IF1^{KO}$, CRL ($n = 7$ each), and $IF1^{TG}$ mice ($n = 8$). Error bars: mean ± SEM. $^*P < 0.05$, $^{**}P < 0.01$, $^{***}P < 0.001$, $^{****}P < 0.0001$ by 1- (H) or 2- (C) way ANOVA with Bonferroni multiple comparisons test, Kruskal–Wallis with Dunn multiple comparisons test (E), or pairwise $t$ test (J and K). See also S2 Fig. Uncropped western blots can be found in S1 Raw Images, and numerical data underlying plots in S1 Data. CRL, control; DG, dentate gyrus; ERK 1/2, extracellular signal-regulated kinase 1/2; fEPSP, field excitatory postsynaptic potential; GluA1, AMPA receptor subunit 1; IF1, ATPase inhibitory factor 1; $IF1^{KO}$, IF1 knockout; $IF1^{TG}$, IF1 overexpressing transgenic; KO, knockout; RhoA, Ras homolog family member A; SC, Schaffer collateral; s.r., stratum radiatum; WAVE1, WASP family member 1.

GluA1 (Fig 7D). MitoQ administration had no significant effect on LTP expression (S3C Fig) or in the levels of NMDA receptor subunits, RhoA, or WAVE1 (S3D Fig).

Importantly, quenching mtROS partially diminished the long-term memory of $IF1^{TG}$ mice, as shown by their inability to recognize the novel object in the long-term memory test (Fig 7E). Of note, MitoQ-treated animals showed high dispersion in the discrimination index for the test session (Fig 7E), although this did not correlate with the redox markers analyzed for some of them (Fig 7B, S3A and S3B Fig), following a random distribution. These changes happened without gross differences in the time exploring both objects (S3E Fig) or in basal exploratory activity (S3F and S3G Fig), suggesting the specificity of mtROS production by the IF1-inbibited ATP synthase in learning and memory.

## IF1 improves exploratory activity, memory, and motor coordination in aged mice

Finally, we studied the exploratory, cognitive, and locomotor behavior in 2-year old mice expressing different doses of IF1. Aged $IF1^{TG}$ mice showed higher exploratory activity when compared to the other 2 genotypes (S4A and S4B Fig). In the long-term memory test, they also exhibited a higher discrimination index in the test session (S4C Fig), with no differences in the participation in the test (S4D Fig), suggesting that they have improved long-term memory. Motor coordination was also better preserved in aged $IF1^{TG}$ mice, especially at low rotating speeds, as assessed in the Rota-rod test (S4E Fig). No significant differences were found in the life span among the 3 genotypes (S4F Fig).

## Discussion

Herein, we show that IF1, an abundant mitochondrial protein in neurons [30], plays a key role in the regulation of metabolism, neuronal function, and learning. Using a genetic approach of loss and gain of function of IF1, we demonstrate that mitochondrial IF1 is bound to and inhibits the synthesis of ATP by a relevant fraction of the enzyme in vivo under physiological conditions. These findings are in agreement with our previous observation in mouse heart mitochondria using a pharmacological approach [28]. Consistently, IF1 co-purifies with and is bound to the tetrameric ATP synthase in porcine and ovine heart mitochondria [26,27] and with monomers and dimers of the enzyme in other mouse tissues [30] and in human cell lines [54].

In vitro, IF1 readily inhibits ATP hydrolysis by the isolated enzyme, and this has been widely assumed to be its function in vivo to prevent reversal of the ATP synthase and wasteful ATP hydrolysis in conditions such as hypoxia [18,19]. However, recent findings in cancer cells cast doubts on the reverse functioning of the ATP synthase in hypoxic conditions, since the

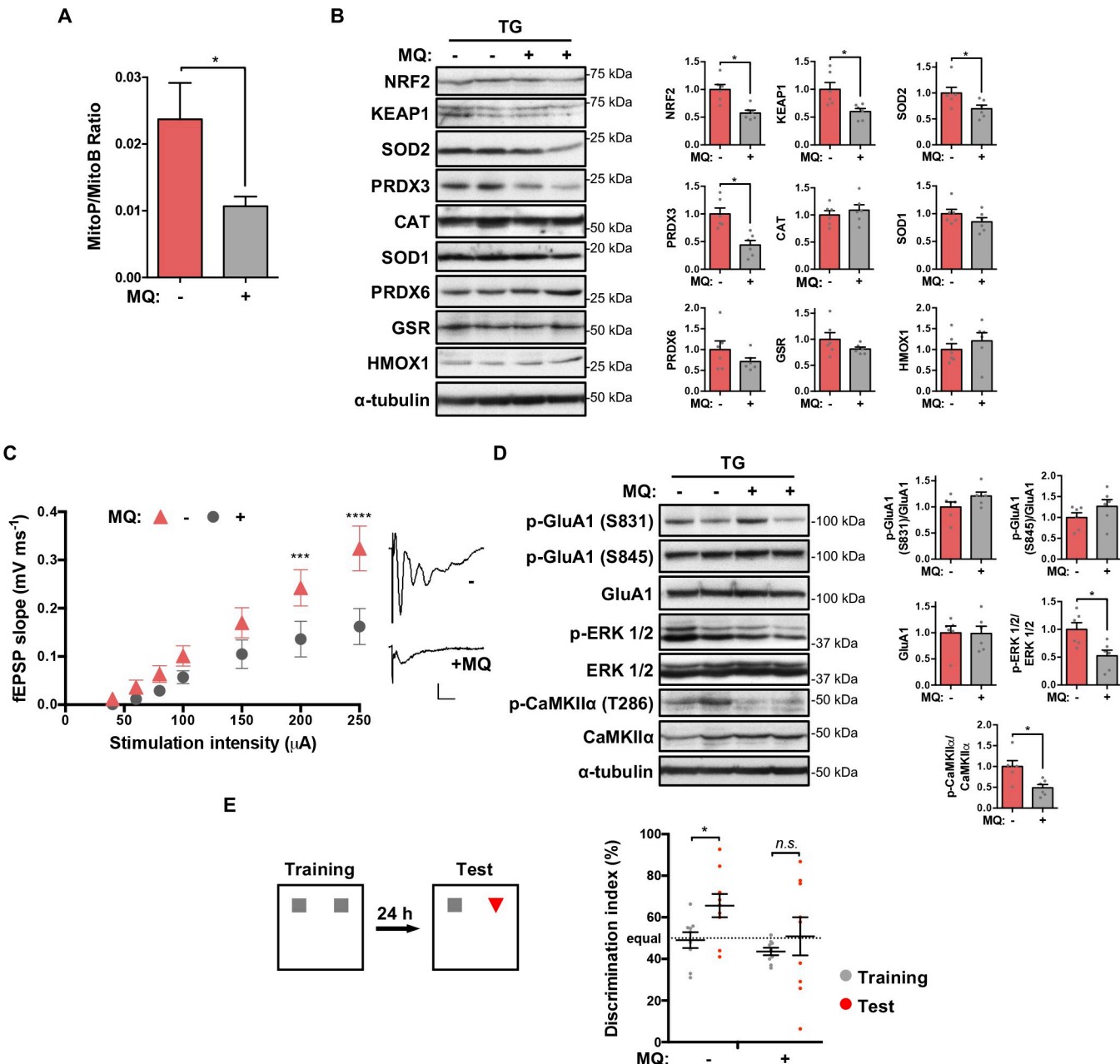

**Fig 7. mtROS signaling mediates the increased synaptic transmission and memory of $IF1^{TG}$ mice. (A)** MitoP/MitoB ratio in the hippocampus of $IF1^{TG}$ untreated ($n = 17$) or MQ-treated mice ($n = 15$). **(B)** Western blots of NRF2, KEAP1, SOD2, PRDX3, CAT, SOD1, PRDX6, GSR, and HMOX1 in hippocampal extracts. α-tubulin is shown as control, and 2 representative samples are shown (1 and 2). Histograms to the right show the quantification as fold of untreated $IF1^{TG}$ mice ($n = 6$). **(C)** Input–output curves showing fEPSP slope in acute hippocampal slices from untreated or MQ-treated $IF1^{TG}$ mice ($n = 11$ slices from 5 mice) in response to increasing stimulus strength. Insets: representative fEPSPs at 250 μA (scale bar, 0.2 mV, 10 ms). **(D)** Western blots of the phosphorylation and expression of GluA1, ERK 1/2, and $Ca^{2+}$/Camk2a. α-tubulin is shown as control. Histograms to the right show the quantification as fold of untreated $IF1^{TG}$ mice ($n = 6$). **(E)** The plot shows the discrimination index in the long-term memory test ($n = 9$). Error bars: mean ± SEM. $^*P < 0.05$, $^{***}P < 0.001$, $^{****}P < 0.0001$ by 2-tailed (A, B, and D) or pairwise (E) $t$ test, or 2-way ANOVA with Bonferroni multiple comparisons test (C). In (C), data from $IF1^{TG}$ mice are replotted from Fig 3, since the recordings were performed in the same experiment. See also S3 Fig. Uncropped western blots can be found in S1 Raw Images, and numerical data underlying plots in S1 Data. Camk2a, Calcium/calmodulin-dependent protein kinase II α; CAT, catalase; ERK 1/2, extracellular signal-regulated kinase 1/2; fEPSP, field excitatory postsynaptic potential; GluA1, AMPA receptor subunit 1; GSR, glutathione reductase; HMOX1, heme oxygenase 1; $IF1^{TG}$, IF1 overexpressing transgenic; KEAP1, kelch-like ECH-associated protein 1; MQ, MitoQ; mtROS, mitochondrial reactive oxygen species; NRF2, nuclear factor erythroid 2-like 2; PRDX3, peroxiredoxin 3; SOD, superoxide dismutase.

enzyme does not hydrolyze ATP under severe hypoxia unless $\Delta\Psi$m is collapsed, a condition mimicking extreme anoxia [55]. Our findings thus support a role for IF1 as an inhibitor of the enzyme in vivo, indicating the coexistence of active and IF1-inhibited ATP synthase pools in neuronal mitochondria. Other recent studies also suggest the coexistence of 2 functional pools of the enzyme in cells [56] or in brain mitochondria [57], supporting the model of $\Delta\Psi$m heterogeneity within the same mitochondrion as functionally independent bioenergetic units [58]. Most importantly, we suggest that the IF1-inhibited ATP synthase slows mitochondrial ATP synthesis leading to an increase in the proton motive and thus in the generation of mtROS for retrograde signaling of nuclear responses [11,20], in this case related to synaptic and cognitive functions. Basal mtROS production by RET is similar among the 3 genotypes. RET at complex I is a relevant signaling pathway operating in vivo during ischemia [32,59], in inflammation [60], life span extension, and as part of the oxygen-sensing mechanism of the carotid body [33]. Since $\Delta\Psi$m controls the level of RET [34], our findings suggest that IF1 dose affects RET by modulating $\Delta\Psi$m in vivo, since $\Delta\Psi$m and mtROS production rate concurrently increase with higher IF1 dose. Perhaps, this signaling phenotype is better observed in the animals used in this study, which are maintained in the C57BL/6J background, which lacks a functional nicotinamide nucleotide transhydrogenase [61]. In any case, we cannot exclude that other signaling molecules emanating from mitochondria could contribute to the phenotypes observed [11,62].

Previous attempts to study the biological function of IF1 by genetic ablation in mice yielded conflicting results. A poorly characterized full *IF1*[KO] mice exhibit no relevant alterations in body weight, breeding, or blood profiles [63]. In contrast, the parental line used to develop our *IF1*[KO] mice in neurons in which IF1 has been globally targeted (the Atpif1[tm1a] mice in Fig 1A), show reduced body weight, fat amount, and circulating levels of alanine transaminase and alkaline phosphatase, suggesting alterations in metabolic regulation (https://www.mousephenotype.org) [64]. Therefore, we support that IF1 is a relevant protein for tissue homeostasis in mammals, being its relevance imposed by its restricted tissue-specific expression [30].

We show that IF1 contributes to defining mitochondrial structure and function in neurons (Fig 8). Although mitochondria are more rounded upon ablation of IF1, what might suggest that the organelles are more fragmented in IF1-KO mice, we observed no differences in DRP1 expression between IF1-KO and CRL mice (Fig 1I) and the aspect ratio of mitochondria—which is an indication of mt-elongation—is significantly higher in CRL mice (Fig 1H). Moreover, calculation of the mitochondrial area is similar among the 3 genotypes (see new Fig 1H). Hence, we rather support that mitochondria are more rounded in IF1-KO mice because of their more disorganized cristae organization. Ablation of IF1 alters cristae organization likely by affecting the oligomerization state of the ATP synthase, which contributes to shaping cristae structure [65]. In fact, IF1 overexpression in mouse liver in vivo promotes the formation of ATP synthase dimers [14], and the increase in the mitochondrial dose of IF1 in neurons increases the oligomeric state of the enzyme. Thus, the lack of IF1 in neurons results in an increase in enzyme activity and in the mitochondrial H$^+$ leak, thus lessening $\Delta\Psi$m and mtROS production and impairing learning, suggesting a role for the IF1-inhibited ATP synthase in cognitive performance (Fig 8). This is in agreement with the recent observation of an increased H$^+$ leak in neurons from fragile X mental retardation 1 (*Fmr1*)-knockout mice, a model of intellectual disability [57]. In contrast, we observed that upon IF1 overexpression, mitochondrial respiratory capacity and glycolytic flux were significantly diminished even though the metabolic sensor AMPK was activated in *IF1*[TG] mice. Perhaps, the better organized cristae structure in mitochondria of *IF1*[TG] mice with less H$^+$ leak might contribute to more efficient energy provision to sustain the higher synaptic transmission in this genotype. Mechanistically,

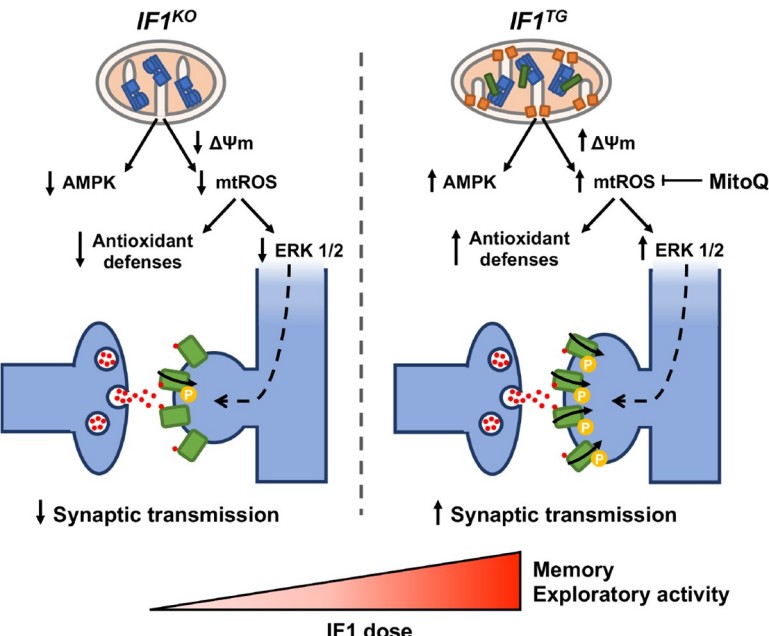

**Fig 8. IF1 promotes synaptic function and cognition in mice.** *IF1^{KO}* in forebrain neurons affects mitochondrial structure and function and impairs memory. IF1 overexpression partially arrests OXPHOS, promotes the activation of AMPK and mtROS/ERK 1/2, and increase synaptic transmission and memory. ΔΨm, mitochondrial membrane potential; AMPK, AMP-activated protein kinase; ERK 1/2, extracellular signal-regulated kinase 1/2; IF1, ATPase inhibitory factor 1; *IF1^{KO}*, IF1 knockout; *IF1^{TG}*, IF1 overexpressing transgenic; mtROS, mitochondrial reactive oxygen species; OXPHOS, oxidative phosphorylation.

increased IF1 expression might reduce glycolytic flux by promoting a more efficient OXPHOS with less dissipation of the proton gradient through the H$^+$ leak, by controlling the activity of glycolytic enzymes through the cellular energy charge. In addition, the reduced dopamine levels found in the hippocampus of *IF1^{TG}* mice might also contribute to their enhanced basal synaptic transmission, since dopamine depresses synaptic transmission in the CA1 region [66]. Furthermore, the partial IF1-mediated decrease of OXPHOS in neurons activates signaling mediated by mtROS and ERK 1/2, which are known to promote increased synaptic transmission and memory [41] (Fig 8). Indeed, growing evidences support that mtROS regulate transmission at both excitatory [67] and inhibitory [68] synapses. Since long-term administration of MitoQ to mice has no effects on whole-body physiology, metabolism, gene expression, or locomotor activity [69], we support the role of mtROS-mediated signaling by the inhibition of the ATP synthase in synaptic and cognitive functions.

Synaptic plasticity is considered the cellular basis for learning and memory, and LTP at CA1 synapses is a prominent mechanism of plasticity [37]. However, its direct involvement in memory is subject of debate [70–72]. In our study, enhanced long-term memory in *IF1^{TG}* mice is better correlated with increased basal transmission and ERK signaling rather than with LTP expression at CA1 synapses. Moreover, we cannot exclude that the activity of other synapses, either in the hippocampus (for example, dentate gyrus to CA3) or in other cortical regions, may be altered with different IF1 dose and could also contribute to the differences in the performance in the novel object recognition task, including the impaired short-term memory of *IF1^{KO}* mice, which show similar basal transmission than controls at CA1 synapses. Novel object recognition is a complex cognitive task [73], and it is unlikely to be fully explained by a single electrophysiological parameter. Moreover, although the core metabolic and redox circuits modulated by IF1 dose are probably conserved across different cell types and brain

regions, the functional synaptic and cognitive outcomes may or may not generalize. For instance, the dentate gyrus-CA3 synapse, which mostly relies on presynaptic, NMDA receptor-independent mechanisms for plasticity [74,75] might be affected in a different way that CA3-CA1 synapse we have studied herein. In any case, we suggest that IF1 inhibition of the ATP synthase contributes to improved cognitive function, perhaps because of the enhanced neuronal activation triggered by ERK signaling in the CA3-CA1 synapse.

Metabolism, redox balance, and neuronal function are controlled by a multipartite coupling between neurons and glial cells, connections that could be affected by mitochondrial IF1 dose. In fact, the metabolism of glutamate and glutamine, which is affected in $IF1^{TG}$ mice, is controlled by a complex shuttle between neurons and astrocytes, which regulates both neuronal metabolism and function [76]. In addition, mtROS produced by astrocytes could also influence neuronal function and learning [77]. Likewise, the participation of microglia, which protect the brain from excessive neuronal activation [78], cannot be ruled out since the transcriptomic analysis suggested a more pro-inflammatory profile in $IF1^{KO}$ mice that might contribute to their learning problems. Hence, we suggest that neuronal IF1 dose might also affect the complex crosstalk among neurons, astrocytes, and microglia.

Defects in the assembly or activity of the ATP synthase may be a driving force for different human neuropathologies [79–81]. Interestingly, IF1 expression is down-regulated in the brain of $Fmr1$-knockout mice [82], in which an aberrant mitochondrial $H^+$ leak has been observed that affects the efficiency of ATP production by OXPHOS, synapse maturation [57], and likely mtROS signaling. The leak occurs through the ATP synthase, and its inhibition attenuates the autistic behaviors in $Fmr1$-KO mice and promotes the maturation of synapses [57], highlighting the pathophysiological relevance of the enzyme in human diseases [4]. Interestingly, overexpression of IF1 in neurons of $IF1^{TG}$ mice also promote the enlargement of dendritic spines, increasing basal synaptic transmission, and long-term memory. Remarkably, J147 is a synthetic drug that inhibits the ATP synthase and reverses the cognitive impairment in mouse models of Alzheimer disease and aging [83].

Overall, this study demonstrates by a genetic approach the role of IF1 as an inhibitor of the mitochondrial ATP synthase under in vivo physiological conditions. Moreover, our work paves the way for future studies to allow deeper dissection of the complex contribution of the ATP synthase/IF1 axis to the regulation of different aspects of neuronal, synaptic, and cognitive functions, which could lead to different outcomes in different brain regions, and highlights its potential as a therapeutic target to treat cognitive deficits associated with neurodegenerative and age-associated pathologies.

## Materials and methods

### Genetically engineered mice

Mouse experiments were carried out after approval of the institutional review board (Ethical Committee of the UAM, CEI-101-1891-A325) in compliance with animal policies and ethical guidelines of the European Community. C57BL/6NTac-Atpif1$^{tm1a(EUCOMM)Wtsi/WtsiCnbc}$ mouse line was acquired from The European Mouse Mutant Archive (EMMA, EM:05233) to develop IF1-floxed mice by breeding with B6;SJL-Tg(ACTFLPe)9205Dym/J mice [84]. Floxed mice were bred with B6.Cg-Tg(Camk2a-cre)T29-1Stl/J mice (The Jackson Laboratory, Bar Harbor, Maine, United States of America; Stock No: 005359) to generate conditional $IF1^{KO}$ mice in forebrain neurons. For inducible overexpression of IF1 in neurons in $IF1^{TG}$ mice, B6. CBA-Tg(tetO-ATPIF1)1Czv mouse line (developed as described in [13]) was bred with B6; CBA-Tg(Camk2a-tetracycline transactivator [tTA])1Mmay/J line [85]. Mice were maintained on C57BL/6J background. The expression of both Cre and transactivator (tTA) transgenes

starts in early postnatal development [86,87], thereby driving the ablation of mouse IF1 or overexpression of human IF1 from that moment. All the mouse experiments were performed, and samples were harvested at 6 to 9 months of age.

$IF1^{TG}$ mice were treated with the mtROS scavenger MitoQ administered in the drinking water at a concentration of 200 μM. The treatment started at 1 month of age and lasted for 9 months for behavioral tests or for 3 months for the determination of mtROS production or for electrophysiological experiments. Drinking of MitoQ-supplemented water was checked on a daily basis.

## PCR genotyping

The primer sequences for genotyping any *Atp5if1* engineered allele (*tm1a*, *floxed* or *KO*; Mut PCR) were 5′-TGCCTGACATTGGTATTGGG-3′ and 5′-TCGTGGTATCGTTATGCGCC-3′, which yielded a 114-bp band. Wild-type and IF1-floxed alleles were distinguished with 5′-TGCCTGACATTGGTATTGGG-3′ and 5′-GTGCAGCTTGTGGGAGTCAG-3′ primers (WT PCR), which yielded 396- and 605-bp bands, respectively. Cre recombinase was detected with 5′-CATTTGGGCCAGCTAAACAT-3′ and 5′-TAAGCAATCCCCAGAAATGC-3′ primers, which yielded a 233-bp band. Regarding $IF1^{TG}$ mice, the primer sequences for genotyping human IF1 were 5′-CACAGAGTAGAGAACAACTG-3′ and 5′-GTTAGTAGCACACAGACAAA-3′, which yielded a 235-bp band, and 5′-CGCTGTGGGGGCATTTTACTTTAG-3′ and 5′-CATGTCCAGATCGAAATCGTC-3′ for tTA, which yielded a 469-bp band.

## Western blot and analysis of protein carbonylation

Tissue pieces were snap frozen after dissection, and proteins were extracted essentially as described [30] using a glass-glass Potter homogenizer in 8 volumes of extraction buffer (50 mM Tris-HCl, pH 8.0, 1% NaCl, 1% Triton X-100, 1 mM DTT, 0.1% SDS, and 0.4 mM EDTA) supplemented with EDTA-free protease inhibitor cocktail (Complete Mini, MilliporeSigma, Burlington, Massachusetts, USA) and phosphatase inhibitor cocktail-2 (MilliporeSigma). Tissue homogenates were freeze-thawed 3 times and centrifuged at 13,000 rpm for 30 minutes at 4°C. Protein concentration was determined with Bradford reagent (Bio-Rad, Hercules, California, USA) using bovine serum albumin (BSA) as standard.

Protein extracts were fractionated by SDS-PAGE, transferred onto nitrocellulose (AmershamProtran 0.2 mm NC, GE Healthcare Life Sciences, Little Chalfont, United Kingdom) or PVDF membranes (Immobilon-P Transfer Membrane, 0.45 μm, MilliporeSigma). Membranes were blocked with 5% nonfat dried milk in Tris-buffered saline (TBS) with 1% Tween 20 for 1 hour at room temperature and incubated with the primary antibody diluted in 3% BSA and 0.05% NaN$_3$ in TBS overnight at 4°C. The following primary antibodies were used: rabbit anti-IF1 [30] (1:1,000), mouse anti-human IF1 (clone 14/2 [22], 1:500), mouse anti-β-F1 (clone 11/21-7A8 [88], 1:1,000), mouse anti-GAPDH (clone 273A-E5 [88], 1:1,000), mouse anti-NADHs9 (NDUFA9, clone 15/22-5 [89], 1:1,000), mouse anti-SDH-B (clone 21A11AE7, Invitrogen, Carlsbad, California, USA, #459230, 1:1,000), mouse anti-UQCRC2 (clone 13G12AF12BB11, Abcam, Cambridge, UK, ab14745, 1:1,000), mouse anti-COX IV (clone 20E8C12, Abcam, ab14744, 1:1,000), rabbit anti-MPC1 (clone D2L9I, CST, Danvers, Massachusetts, USA, #14462, 1:1,000), mouse anti-HSP60 (clone 17/9-15 G1 [88], 1:5,000), mouse anti-OPA1 (clone 18/OPA1, BD Biosciences, San Jose, California, USA, #612606, 1:1,000), mouse anti-MFN2 (clone XX-1, SCBT, Dallas, Texas, USA, sc-100560, 1:1,000), mouse anti-DLP1 (DRP1, clone 18/DLP1, BD Biosciences, #611112, 1:1,000), rabbit anti-VDAC1 (Abcam, ab15895, 1:500), mouse anti-IMMT (clone B-10, SCBT, sc-390706, 1:1,000), rabbit anti-AMPKα (clone D5A2, CST, #5831, 1:1,000), rabbit anti-p-AMPKα (CST, #2531, 1:1,000),

rabbit anti-AMPKβ 1/2 (clone 57C12, CST, #4150, 1:1,000), rabbit anti-p-AMPKβ 1 (CST, #4181, 1:1,000), rabbit anti-ACC (CST, #3662, 1:1,000), rabbit anti-p-ACC (CST, #3661, 1:1,000), mouse anti-PYGM (Abcam, ab88078, 1:1,000), mouse anti-LDHA (clone 4D3-A1 [89], 1:1,000), mouse anti-α-tubulin (clone DM1A, Sigma, T9026, 1:10,000), mouse anti-CAT (clone 2A1, Invitrogen, #LF-MA0004, 1:500), rabbit anti-SOD1 (SCBT, sc-11407, 1:1,000), rabbit anti-SOD2 (SCBT, sc-30080, 1:1,000), rabbit anti-PRDX3 (Abcam, ab222807, 1:1,000), rabbit anti-PRDX6 (Abcam, ab59543, 1:1,000), rabbit anti-NRF2 (SCBT, sc-722, 1:1,000), rabbit anti-KEAP1 (Sigma, HPA005558, 1:250), rabbit anti-GCLC (Abcam, ab53179, 1:500), mouse anti-GSR (clone C-10, SCBT, sc-133245, 1:1,000), rabbit anti-HMOX1 (clone EPR1390Y, Abcam, ab68477, 1:1,000), rabbit anti-GluA1 (Abcam, ab31232, 1:1,000), rabbit anti-p-GluA1 S831 (Sigma, AB5847, 1:500), rabbit anti-p-GluA1 S845 (Invitrogen, OPA1-04118, 1:1,000), rabbit anti-GluN1 (clone 1.17.2.6, Sigma, AB9864, 1:1,000), rabbit anti-GluN2A (Sigma, 07–632, 1:500), mouse anti-GluN2B (clone 59/20, NeuroMab, Davis, California, USA, 75–097, 1:500), mouse anti-ERK 1/2 (clone C-9, SCBT, sc-514302, 1:1,000), rabbit anti-p-ERK 1/2 (CST, #9101, 1:1,000), rabbit anti-Camk2a (Sigma, C6974, 1:5,000), mouse anti-p-Camk2a (clone 22B1, Sigma, 05–533, 1:2,000), mouse anti-PSD-95 (clone K28/43, NeuroMab, 75–028, 1:1,000), mouse anti-RhoA (clone 26C4, SCBT, sc-418, 1:1,000), mouse anti-WAVE1 (clone E-2, SCBT, sc-271507, 1:1,000), mouse anti-β-actin (clone AC-74, Sigma, A5316, 1:10,000), and mouse anti-3-nitrotyrosine (clone 39B6, Abcam, ab61392, 1:1,000). Peroxidase-conjugated anti-mouse or anti-rabbit IgGs (1:3,000, Nordic Immunology, Rangeerweg, the Netherlands) diluted in 5% nonfat dried milk in TBS with 1% Tween 20 were used as secondary antibodies. The Novex Enhanced Chemiluminescence (Thermo Fisher Scientific, Waltham, Massachusetts, USA) system was used to visualize the bands. The intensity of the bands was quantified with a GS-900 Calibrated Densitometer (Bio-Rad) and ImageJ software version 2.1.0/1.53c (National Institutes of Health, Bethesda, Maryland, USA).

The carbonylation of proteins was detected with the OxyBlot Protein Oxidation Detection Kit (MilliporeSigma) according to manufacturer's instructions.

## Mitochondrial isolation and determination of enzymatic activities

Mitochondria were isolated from fresh forebrain, which was minced and homogenized in a glass-glass homogenizer with 10 volumes of buffer A (320 mM sucrose, 1 mM EDTA and 10 mM Tris-HCl, pH 7.4). Nuclei and unbroken cells were removed by centrifugation at 800 ×$g$ for 10 minutes at 4˚C, and mitochondria were pelleted by centrifugation at 7,500 ×$g$. The ATP synthetic activity was determined in fresh mitochondria as described [28,90], using respiration buffer (225 mM sucrose, 10 mM KCl, 5 mM MgCl$_2$, 1 mM EGTA, 0.1% fatty acid-free BSA, 10 mM K$_2$HPO$_4$ and 10 mM Tris-HCl, pH 7.4) with 120 μM P$^1$,P$^5$-di(adenosine-5′)pentaphosphate, 5 mM ADP and 8 mM succinate as respiratory substrate. The ATP hydrolase, rotenone-sensitive NADH dehydrogenase (complex I), and cytochrome $c$ oxidase (complex IV) activities were determined after freeze-thawing mitochondrial preparations as described [12,91]. A total of 140-μM phenazine methosulphate was used as electron acceptor for the determination of complex I activity. Inhibition of the ATP synthase and hydrolase activities was accomplished by the addition of 10-μM oligomycin to mitochondria from the 3 genotypes.

## Co-immunoprecipitation assays

Co-immunoprecipitation was performed essentially as described [28]. Isolated mitochondria were lysed in 50 mM Tris-HCl, pH 6.0, 150 mM NaCl, 0.5% Nonidet P40, 0.5% Triton X-100, complete EDTA-free protease inhibitor cocktail (Roche, Mannheim, Germany) and

phosphatase inhibitor cocktail (MilliporeSigma). Protein from lysates (300 μg) was incubated with 25 μl of anti-$F_1$-ATPase antibody [92] overnight at 4˚C and bound to EZ View Red Protein G Affinity Gel (MilliporeSigma) for 2 hours at 4˚C. β-F1 was detected with an anti-β-F1 antibody [88] (1:500) labeled with Cy5 using Amersham CyDye Reactive Dye Pack (GE Healthcare), and IF1 was detected with rabbit anti-IF1 [30] (1:1,000).

## Blue native gels

Mitochondrial pellets were solubilized with digitonin at 4:1 g protein and fractionated in blue native (BN) gels [28]. A total of 30 μg of mitochondrial proteins were fractionated in Native-PAGE 3–12% Bis-Tris Protein Gels, 1.0 mm, 10-well (Invitrogen) at 4˚C using cathode (50 mM Tricine, 15 mM Bis-Tris, pH 7.0 and 0.02% Serva blue G) and anode buffers (50 mM Bis-Tris, pH 7.0). Electrophoresis was carried out at a constant voltage (70 V) until the samples entered the polyacrylamide gradient (approximately 30 minutes) and then at a constant current (15 mA) until the dye reached the end of the gel (approximately 1 hour). After fractionation, the gels were electroblotted onto PVDF membranes and processed for immunoblot analysis as described above. The following primary antibodies were used: mouse anti-NADHs9 (NDUFA9, clone 15/22-5 [89], 1:1,000), mouse anti-NDUFS3 (clone 17D95, Abcam, ab14711, 1:1,000), mouse anti-NDUFS4 (clone 2C7CD4AG3, Abcam, ab87399, 1:1,000), mouse anti-SDH-A (clone 2E3GC12FB2AE2, Abcam, ab14715, 1:1,000), mouse anti-UQCRC2 (clone 13G12AF12BB11, Abcam, ab14745, 1:1,000), mouse anti-MT-CO1 (clone 1D6E1A8, Invitrogen, #459600, 1:1,000), mouse anti-MT-CO3 (clone DA5BC4, Abcam, ab110259, 1:1,000), mouse anti-COX IV (clone 20E8C12, Abcam, ab14744, 1:1,000), and rabbit anti-β-F1 [93] (1:20,000).

## Electron microscopy

Animals were transcardially perfused with 4% paraformaldehyde, 2% glutaraldehyde in 0.1 M phosphate buffer, pH 7.4, and the brain was dissected and leaved immersed in the fixative overnight at 4˚C. Moreover, 200-μm width coronal slices were obtained in a vibratome (Leica VT1200S, Wetzlar, Germany), postfixed with a mixture of 2% osmium tetroxide and 1% potassium ferricyanide in $ddH_2O$ for 1 hour at 4˚C, and washed 3 times 10 minutes each in $ddH_2O$. Then, the tissue was treated with 2% uranyl acetate for 1 hour at room temperature, washed again, dehydrated in increasing series of ethanol, and infiltrated in TAAB 812 epoxy resin with the help of propylene oxide. Resin was polymerized for 2 days at 60˚C. Ultrathin sections (70 to 80 nm) containing the hippocampus were obtained using an Ultracut E ultramicrotome, stained with 2% uranyl acetate for 7 minutes and with lead citrate for 2 minutes, and examined at 80Kv in a Jeol JEM-1010 transmission electron microscope coupled to a digital camera Tem-Cam-F416 (4Kx4K) (TVIPS, Gauting, Germany). Samples were processed at the CBMSO Electron Microscopy Facility. Mitochondrial shape descriptors and cristae measurements were obtained manually in ImageJ [94,95].

## Determination of mtDNA copy number

Total genomic DNA (nuclear and mitochondrial) was extracted from the forebrain with phenol:chloforofom:isoamyl alcohol. Mitochondrial/nuclear DNA ratio was quantified with Fast SYBR Master Mix in an ABI PRISM 7900HT sequence detection system (Thermo Fisher Scientific) from the Genomics and NGS Core Facility (CBMSO). Thermal cycling conditions were as follows: initial denaturation of 20 seconds at 95˚C, 40 amplification cycles of 1 second at 95˚C, and 20 seconds at 60˚C each, followed by a dissociation curve analysis to detect possible nonspecific amplification. Standard curves with serial dilutions of pooled DNA were used

to assess amplification efficiency of the primers and to establish the dynamic range of DNA concentration for amplification, which was 10 ng per run. The relative copy number of mtDNA molecules was determined with the comparative $\Delta\Delta C_T$ method using *Sdha* as a nuclear gene and *mt-12S* as a mitochondrial gene. Primers used for amplification were as follows: Sdha (5′-TACTACAGCCCCAAGTCT-3′ and 5′-TGGACCCATCTTCTATGC-3′) and mt-12S (5′-AAACAGCTTTTAACCATTGTAGGC-3′ and 5′-TTGAGCTTGAACGCTTTC TTTA-3′) (Sigma).

## Determination of ATP and ADP

ATP and ADP were determined in flash-frozen total forebrain pieces after homogenization in 6 volumes of 6% perchloric acid and neutralization with KOH, using the ATP Bioluminescence Assay Kit CLS II (Roche) and ADP Assay Kit (Sigma), respectively.

## Primary cultures of mouse hippocampal neurons

Primary neuronal cultures were prepared as described [12]. Breedings were set between heterozygous Atpif1-KO mice (Atpif1^lox/+ Cre) to obtain CRL (Atpif1^+/+ Cre) and *IF1^KO* (Atpif1^lox/lox Cre) embryos and between tetO-IF1 (I^+) and Camk2a-tTA (T^+) to obtain control (wild type or T^+) and double-transgenic (I^+/T^+) embryos. Hippocampi were dissected from E17 to E18 embryos, digested (0.4 mg/ml papain, Roche) and mechanically dissociated, and cells were seeded in serum-free B27-supplemented Neurobasal (Gibco, Grand Island, New York, USA) medium. Cells were plated in poly-D-lysine (10 μg/ml for plastic surface and 50 μg/ml for glass surface) and laminin-coated (1:1,000, Sigma) pretreated plates in medium containing 20% horse serum (Gibco) for 3 hours. Then, medium was completely replaced by serum-free Neurobasal medium supplemented with 2% B27, 1% GlutaMAX (all from Gibco), and 100 U/ ml penicillin-streptomycin. Culture medium was partially replaced every 2 days, and experiments were performed at 14 to 15 days in vitro (DIV).

## Analysis of cellular respiration and glycolytic activity

Oxygen consumption rates (OCRs) and ECARs were determined with Seahorse XF24 Extracellular Flux Analyzer (Agilent Technologies, Santa Clara, California, USA). Primary hippocampal neurons were plated at a density of 25,000 cells/well and cultured for 14 DIV. Cells were equilibrated with Seahorse XF base medium (Agilent Technologies) supplemented with 2.5 mM glucose and 2 mM $CaCl_2$ for 1 hour previous to assay at 37°C in a $CO_2$-free incubator. Mitochondrial function was determined through sequential addition of 6 μM oligomycin, 1 μM FCCP (carbonyl cyanide-4-(trifluoromethoxy)phenylhydrazone), 1 μM antimycin A, and 1 μM rotenone.

The rates of extracellular lactate production were determined using the lactate dehydrogenase-catalyzed oxidation of lactate into pyruvate, by measuring NADH production [22]. Respiratory and glycolytic rates were normalized by protein amount per well, after extraction with cellular lysis buffer (10 mM Tris, pH 7.5, 130 mM NaCl and 0.5% Triton X-100).

## Measurement of ΔΨm and mtROS

ΔΨm and mtROS were determined in neuron cultures by staining with 100 nM TMRM and 2.5 μM MitoSOX probes (Invitrogen), respectively [12]. In the TMRM staining, the nonpotentiometric probe MitoTracker Green FM (100 nM, Invitrogen) was also included to normalize TMRM signal by the signal of mitochondrial mass. Images were acquired in an Axioskop2 Plus (Zeiss, Oberkochen, Germany) with Lambda 10–2 Optical Filter Changer (Sutter

Instrument, Novato, California, USA) coupled to an ORCA-Flash4.0 LT sCMOS camera (Hamamatsu, Hamamatsu, Japan) from Optical and Confocal Microscopy Facility at CBMSO. Specificity of TMRM staining was assessed by the addition of 5 μM oligomycin or 5 μM FCCP to collapse ΔΨm.

## Measurement of $H_2O_2$ production

The production and efflux of $H_2O_2$ was determined in isolated forebrain mitochondria using a plate reader fluorimeter (FLUOstar Omega, BMG Labtech, Ortenberg, Germany) as previously reported [34]. In brief, freshly isolated forebrain mitochondria (100-μg protein) were incubated with 2.5 μM Amplex Red (Invitrogen) and 5 U/ml horseradish peroxidase in respiration buffer with 10 mM succinate at 37˚C. Resorufin (the product of Amplex Red oxidation) fluorescence was detected using excitation and emission filters at 570 and 585 nm, respectively, for 5 minutes (the rates were linear over 10 minutes). FCCP (5 μM) or rotenone (1 μM) were added 30 seconds before measurement.

## Gene array hybridization

The transcriptomic analysis was performed by hybridization of total RNA extracted from the hippocampus of *IF1*^*KO*^, control, and *IF1*^*TG*^ mice (*n* = 3) onto mouse gene arrays (GE 4x44K v2 Microarray Kit, G4846A, Agilent Technologies), as previously described [96]. The hybridizations and statistical analyses were performed by the Genomics Facility at Centro Nacional de Biotecnología (Madrid, Spain). Differentially expressed genes were selected with a fold change ≥1.5 or ≤ –1.5 and a *P* value (LIMMA) < 0.05. PCA was performed using *R* package by Genomics and NGS Core Facility at CBMSO. Genecodis3 [97] and GSEA [98] tools were used to perform GSEA to infer the main biological processes and cellular components associated with the differential expression of IF1. IPA (Qiagen, Germantown, Maryland, USA) was also used to infer the canonical pathways and biological functions significantly enriched in the gene list and predict their activation state. A Fisher exact right-tailed test identified significantly enriched pathways, and a *z* score was computed to predict whether the pathways were activated or inhibited in each comparison.

## RT-PCR analysis

Reverse transcription reactions were performed using 1 μg of total RNA and the High Capacity cDNA Reverse Transcription Kit (Thermo Fisher Scientific). Real-time PCR was done with Fast SYBR Master Mix in an ABI PRISM 7900HT sequence detection system (Thermo Fisher Scientific) from the Genomics and NGS Core Facility (CBMSO). Thermal cycling conditions were as follows: initial denaturation of 20 seconds at 95˚C, 40 amplification cycles of 1 second at 95˚C, and 20 seconds at 60˚C each, followed by a dissociation curve analysis. Standard curves with serial dilutions of pooled cDNA were used to assess amplification efficiency of the primers and to establish the dynamic range of cDNA concentration for amplification, which was 3 ng of input RNA per run. The relative expression of the mRNAs was determined with the comparative $\Delta\Delta C_T$ method using GAPDH and β-actin as controls. Primers used to amplify the target genes were as follows: GAPDH (5′-TGCGACTTCAACAGCAACTC-3′ and 5′-GGATAGGGCCTCTCTTGCTC-3′) and β-actin (5′-AACACAGTGCTGTCTGGTGGT-3′ and 5′-GATCCACATCTGCTGGAAGG-3′) (Sigma). KiCqStart SYBR Green Primers (Sigma) were used for the confirmation of the gene array results of mouse *Camk1d* (Gene ID 227541, primer pair 1), *Camk2b* (Gene ID 12323, primer pair 1), *Mapk1* (Gene ID 26413, primer pair 1), *Map2k4* (Gene ID 26398, primer pair 1), *Snca* (Gene ID 20617, primer pair 1), *Ikbkap* (Gene ID 230233, primer pair 2), *Tia1* (Gene ID 21841, primer pair 1), *Gps2* (Gene ID 56310,

primer pair 1), *Actg1* (Gene ID 11465, primer pair 1), *Dld* (Gene ID 13382, primer pair 1), *Grin1* (Gene ID 14810, primer pair 1), and *Grin2b* (Gene ID 14812, primer pair 1).

## Proteomic analysis

The quantitative proteomic analysis of total hippocampal extracts of $IF1^{KO}$, control, and $IF1^{TG}$ mice ($n$ = 2) was performed with isobaric tags for relative and absolute quantitation (iTRAQ) labeling method coupled to tandem mass spectrometry (MS/MS) as previously described [96]. In situ digestion of protein extracts, iTRAQ labeling, reverse phase-liquid chromatography-MS/MS analysis, and protein identification and quantitation were performed by the CBMSO Proteomic Facility.

Protein identification from raw data was carried out using the SEQUEST algorithm (Proteome Discoverer 1.4, Thermo Fisher Scientific) and PEAKS 8 Studio (Bioinformatics Solutions, Waterloo, Canada). Database search was performed against the mouse reference proteome (UniProt) with the following constraints: tryptic cleavage after Arg and Lys, up to 2 missed cleavage sites, tolerances of 20 ppm for precursor ions and 0.05 Da for MS/MS fragment ions, and the searches were performed allowing optional Met oxidation, Cys carbamidomethylation, and iTRAQ reagent labeling at the N-terminus of Lys residues. Search was performed against decoy database (integrated decoy approach) using false discovery rate (FDR) < 0.01. All proteins were identified with at least 2 high confidence peptides. Quantitation of iTRAQ labeled peptides was performed with Proteome Discoverer 1.4 and PEAKS 8 software using ANOVA test and a significance threshold of $\geq 7$. Enrichment analyses were performed as described above for the transcriptome.

## Metabolomic analysis

Amino acids and neurotransmitters were extracted from flash-frozen hippocampi by homogenization in 8 volumes of 1.3% perchloric acid with 0.5 mM EDTA in glass tubes. After 10-minute incubation on ice, proteins were pelleted at 13,000 rpm for 5 minutes. Hippocampal amino acids were analyzed by ultra-high-performance liquid chromatography (UPLC) coupled to MS/MS, as previously reported [99]. Briefly, amino acids were separated in a Waters ACQUITY UPLC H-class chromatograph and quantified with a Waters Xevo TQD triple-quadrupole mass spectrometer using positive electrospray ionization conditions in the MRM mode. Standard curves of analytical standard solutions of acid and neutral and basic amino acids (MilliporeSigma) were prepared. All the samples and standard curve points were spiked with 1 nmol L-norleucine (MilliporeSigma) as an internal standard.

Neurotransmitters were analyzed by ion pair HPLC separation (1500 Series, Waters) coupled to electrochemical detection (Coulochem II, Hucoa-Erlöss SA, Madrid, Spain) [100]. Standard curves of 3-o-methyldopa, 4-hydroxy-3-methoxyphenylglycol, 5-hydroxytryptophan, 5-hydroxyindoleacetic acid, and homovanillic acid (Sigma) were prepared, and a titrated internal control based on lyophilized urine (Especial Assays Urine, MCA Laboratory, Winterswijk, the Netherlands) was used for spiking.

Organic acids were extracted with ethyl acetate and diethyl ether (Sigma) and derivatized with bis(trimethyl-silyl)trifluoro-acetamide (BSTFA) (Sigma) [101]. The trimethylsilyl derivatives obtained were separated by gas chromatography (Agilent 7890A) and detected in a mass spectrometer (Agilent 5975C). Standard curves were prepared with ERNDIM Organic Acids mixture (MCA Laboratory), and undecanedioic acid (Sigma) was used for spiking.

Determination of tissue GSH was carried out using the Glutathione Fluorometric Assay Kit (BioVision, Milpitas, California, USA).

Metabolic pathway analysis was performed using MetaboAnalyst 4.0 [102] using Global test pathway enrichment analysis and relative betweenness centrality for pathway topology analysis, searching against mouse Kyoto Encyclopedia of Genes and Genomes (KEGG) pathway library.

## Electrophysiology

Excitatory transmission was recorded from synapses between Schaffer collaterals and CA1 neurons in the hippocampus with field recordings [38]. Acute brain slices were prepared from 8 to 9 months old mice. Animals were anaesthetized with isoflurane and decapitated. The brain was rapidly removed and placed in ice-cold dissection medium (10 mM glucose, 4 mM KCl, 26 mM NaHCO$_3$, 233.7 mM sucrose, 5 mM MgCl$_2$ and phenol red as a pH indicator) gassed with 95% O$_2$/5% CO$_2$ to keep pH at 7.3. Moreover, 300-μm thick coronal slices containing the hippocampus were sectioned with a vibratome. Slices were incubated in artificial cerebrospinal fluid (ACSF: 119 mM NaCl, 2.5 mM KCl, 1 mM NaH$_2$PO$_4$, 26 mM NaHCO$_3$, 1.2 mM MgCl$_2$, 2.5 mM CaCl$_2$, 11 mM glucose, and gassed with 95% O$_2$/5% CO$_2$) for 1 hour at 32˚C and then at room temperature.

Field recordings were performed in a recording chamber that was continuously perfused with ACSF constantly gassed with 95% O$_2$/5% CO$_2$ and kept at 25˚C. ACSF was supplemented with 100 μM picrotoxin, an antagonist of gamma-aminobutyric acid class A receptors (GABA$_A$), to block inhibitory neurotransmission. fEPSPs were recorded with a low-resistance (0.4 to 1.0 MΩ) glass electrode filled with ACSF, placed in CA1 stratum radiatum. The responses were evoked with bipolar stimulation electrodes consisting of platinum/iridium cluster electrodes of 25 μm in diameter (FHC, Bowdoin, Maine, USA). They were placed over Schaffer collateral fibers between approximately 300 and 500 μm away from the recorded field with visual guidance using transmitted light illumination in a microscope equipped with a 10× objective (Olympus BX50WI microscope, Shinjuku, Japan). Recordings were obtained with Multiclamp 700 A/B amplifiers and pClamp software (Molecular Devices, San Jose, California, USA). All the components were placed on an air table to absorb mechanical vibrations and inside a Faraday cage to block interference from electrical noise.

fEPSPs were recorded at different stimulation intensities for each slice to generate an input–output curve to determine basal synaptic transmission. This curve was also used to set the baseline fEPSP value at about 30% of the maximum for LTP experiments. Baseline stimulation was delivered every 15 seconds (0.01-ms pulses) for at least 20 minutes before LTP induction to ensure the stability of the response. LTP in the CA1 was induced by a theta burst stimulation (bursts of 4 pulses at 100 Hz, with the bursts repeated in 10 trains at 5 Hz, and the trains repeated 4 times separated by 15 seconds), and the responses were recorded for 1 hour after induction. Synaptic plasticity changes were determined as the normalized change in average response during the last 5 minutes of recording (55 to 60 minutes after induction protocol) compared to the 20-minute baseline. To illustrate the time course, synaptic parameters were grouped in 1.5-minute bins.

## Transfection of primary neuronal cultures and spine analysis

Cultures in 24-well plates (100,000 cells/well) were transfected at 12 DIV using 1-μl Lipofectamine 2000 (Invitrogen) with 1 μg pEGFP-N3 for the visualization of dendrites and spines. Moreover, 24 to 48 hours post-transfection, cells were fixed with 2% paraformaldehyde in phosphate-buffered saline for 20 minutes at 37˚C. Images were obtained in a confocal microscope A1R+ (Nikon, Minato, Japan) from Optical and Confocal Microscopy Facility at CBMSO using a 60×/1.4 (NA) Plan apochromatic oil immersion objective. Series of images

were acquired in the *z* dimension and processed with ImageJ for manual analysis of dendritic spines [103].

## Behavioral tests

Adult (8 to 10 months old) or old (22 to 24 months old) male mice were used in the behavioral tests. Animals were handled for 5 minutes in the experimental room 3 days prior to beginning the experiments for habituation purposes. To avoid the influence of circadian fluctuations, the behavioral tests were carried out during the light period, from 10 AM to 6 PM. All the tests were recorded for their analysis using ANY-maze software (Stoelting, Wood Dale, Illinois, USA).

In the open field test, mice freely explored for 5 minutes a $40 \times 40 \times 35$ cm (width $\times$ height $\times$ depth) open field arena made of transparent plastic walls and black plastic bottom. Wood shaving bedding was put on the arena floor and changed for each mouse. In ANY-maze, the arena was divided with a virtual grid of $10 \times 10$ cm squares and total distance traveled and number of lines of the virtual grid crossed, and time mobile were measured.

Memory was assessed with the novel object recognition test [49]. In brief, this test consisted in one training session, in which mice explored the open field arena described above with 2 identical objects (sample objects) for 10 minutes and then returned to their home cages. Two or 24 hours later, mice were subjected to the test session to analyze short- or long-term memory, respectively, exploring the same arena for 10 minutes, but one object was changed by a novel one. The preference for the novel object, which depends on the ability to recognize it, was determined as a discrimination index, calculated as the time interacting with the novel object with respect to the total time exploring both objects. The discrimination index of the training sessions was also calculated to control inherent preferences for any particular object or positioning. Mice showing a discrimination index deviating more than 20% from the expected 50% in the training session were excluded.

Scopolamine (0.3 mg/kg) was administered via intraperitoneal injection in saline solution right after the training session or 30 minutes before the test session to interfere with memory consolidation or recall processes, respectively.

Motor balance and coordination were assessed in old mice using an accelerating rotarod (UGO Basile Accelerating Rotarod, Gemonio, Italy). Mice first received a training session consisting of 5 runs of 1 minute each, walking at a low speed (4 rpm). The day after, mice had to run at increasing speeds for 1 minute each. Each mouse had 5 trials to run for 1 minute at each speed without falling. If it failed in the 5 trials, the test finished. The average latency to fall was calculated for each genotype for each speed.

## In vivo determination of the production of mtROS

The MitoB/MitoP exomarker method [53] was used to assess the production of mtROS. Moreover, 2.5-nmol MitoB in 0.2 μl of saline solution were stereotaxically injected into the hippocampus in anesthetized mice. Coordinates (mm) were anteroposterior (AP; from bregma) 2.0, medio-lateral (ML) 1.5, and dorsoventral (DV; from dura) 2.0. Mice were humanely killed 4 hours later, and the hippocampus was weighed and flash-frozen at $-80°C$. MitoB and MitoP were extracted with 60% acetonitrile/0.1% formic acid, and standard curves of MitoB and MitoP were prepared and spiked with the internal standards $d_{15}$-MitoB and $d_{15}$-MitoP. The MitoB and MitoP content was measured by liquid chromatography with tandem mass spectrometry (LC–MS/MS) [53].

## Statistical analysis

The results shown are the mean ± SEM. Statistical analyses were performed by Student $t$ test, 1- or 2-way ANOVA with a post hoc Bonferroni test, or Kruskal–Wallis test with a post hoc Dunn test as appropriate. Survival curves were derived from Kaplan–Meier estimates and compared by log-rank test. Statistics were calculated using GraphPad Prism version 6. Values of $P < 0.05$ were considered statistically significant. Statistical details and methods used in each experiment can be found in the figure legends. $P$ values are provided in figure legends (*$P < 0.05$, ** $P < 0.01$, *** $P < 0.001$, **** $P < 0.0001$).

## Supporting information

**S1 Fig. IF1 dose does not affect the expression and assembly of proteins of the respiratory chain. (A)** Representative western blot of IF1 and β-F1 expression in different brain regions and tissues from CRL (C) and $IF1^{KO}$ mice. **(B)** Representative western blot of IF1 in forebrain extracts from CRL (+/+), heterozygous (+/−) and $IF1^{KO}$ (−/−) mice. Two independent samples are shown (1 and 2). The histogram to the left shows the quantification of IF1/β-F1 ratio ($n = 6$–8). There is some IF1 expression in $IF1^{KO}$ mice because the Cre recombinase is expressed in excitatory neurons, hence other neuronal types have not undergone recombination. **(C)** Western blot analysis of the expression of OXPHOS complexes I (NADHs9), II (SDH-B), III (UQCRC2), IV (COX IV), ATP synthase (β-F1), IF1, and MPC1 in hippocampal extracts of $IF1^{KO}$, CRL, and $IF1^{TG}$ mice. α-tubulin is shown as control, and 2 representative samples are shown (1 and 2). Histograms to the right show the quantification as fold change of CRL ($n = 5$). **(D)** Representative BN immunoblots probed with antibodies against mitochondrial complexes I (NDUFA9, NDUFS3 and NDUFS4), II (SDH-A), III (UQCRC2), IV (COX IV, MT-CO1 and MT-CO3), and ATP synthase (β-F1). The migration of oligomers $(F_oF_1)_n$, dimers $(F_oF_1)_2$ and monomers $(F_oF_1)$ of the ATP synthase and other superassemblies is indicated. VDAC is shown as control. Histograms to the bottom show the quantification of ATP synthase monomers and superassemblies (dimers and oligomers) as fold change of CRL ($n = 3$). **(E)** The histograms show the rate of $H_2O_2$ production in isolated forebrain mitochondria using succinate as respiratory substrate in the absence or presence of FCCP or rotenone ($n = 3$–4). Error bars: mean ± SEM. *$P < 0.05$, **$P < 0.01$ by 2-tailed $t$ test (B–E). Related to Fig 2. Uncropped western blots can be found in S1 Raw Images, and numerical data underlying plots in S1 Data. BN, blue native; CRL, control; IF1, ATPase inhibitory factor 1; $IF1^{KO}$, IF1 knockout; $IF1^{TG}$, IF1 overexpressing transgenic; MPC1, mitochondrial pyruvate carrier 1; OXPHOS, oxidative phosphorylation; VDAC, voltage-dependent anion channel. (TIF)

**S2 Fig. IF1 overexpression enhances long-term memory. (A)** Time course of the relative changes in fEPSP slope before and after TBS in hippocampal slices from $IF1^{KO}$ and CRL ($n = 6$–7 slices from 4 mice) and $IF1^{TG}$ mice ($n = 3$ slices from 3 mice). Insets: representative fEPSPs before (gray line) and after (black) TBS. The histogram to the right summarizes the last 5 minutes of the time courses shown in (A). **(B)** Western blots of the phosphorylation and expression of NMDA receptor subunits 2A, 2B, and 1 (GluN2A, GluN2B, and GluN1), calcium/Camk2a and PSD-95. α-tubulin is shown as control, and 2 representative samples are shown (1 and 2). Histograms to the right show the quantification as fold of CRL ($n = 5$). **(C)** qPCR analysis of the mRNA levels of NMDA receptor subunits 1 (Grin1) and 2B (Grin2b) in $IF1^{KO}$, CRL, and $IF1^{TG}$ mice ($n = 6$). **(D)** Total time spent exploring both objects in the training or test sessions of the short- (left) and long-term (right) memory tests for $IF1^{KO}$, CRL ($n = 7$ each), and $IF1^{TG}$ mice ($n = 8$). **(E)** Long-term memory tests with Scop administration.

(F and G) Box plots show the discrimination indexes when Scop was injected after the training (**F**) or before the test session (**G**) to $IF1^{TG}$ mice ($n = 8$). Error bars: mean ± SEM. $^*P < 0.05$, $^{**}P < 0.01$ by 2-tailed (--D) or pairwise (F and G) $t$ tests, or Kruskal–Wallis followed by Dunn multiple comparisons test (A); Wilcoxon test was used to analyze LTP expression with respect to baseline (A). Related to Fig 6. Uncropped western blots can be found in S1 Raw Images, and numerical data underlying plots in S1 Data. Camk2a, Calcium/calmodulin-dependent protein kinase II α; CRL, control; fEPSP, field excitatory postsynaptic potential; IF1, ATPase inhibitory factor 1; $IF1^{KO}$, IF1 knockout; $IF1^{TG}$, IF1 overexpressing transgenic; PSD-95, postsynaptic density protein 95; qPCR, quantitative polymerase chain reaction; Scop, scopolamine; TBS, theta burst stimulation.
(TIF)

**S3 Fig. MQ treatment diminishes mtROS-mediated signaling in $IF1^{TG}$ mice. (A and B)** Representative blots of the extent of carbonylation (A) and tyrosine nitration (B) of hippocampal proteins. α-tubulin is shown as control. Arrowheads to the left of the blots identify the migration of the proteins used in the quantification of carbonylation and nitration (histograms to the bottom, $n = 6$). **(C)** Time course of relative changes in fEPSP slope before and after TBS in hippocampal slices from $IF1^{TG}$ mice untreated ($n = 3$ slices from 3 mice) or treated with MQ ($n = 4$ slices from 4 mice). Insets: representative fEPSPs before (gray line) and after (black) TBS. The histogram to the right summarizes the results from the last 5 minutes of the time courses. **(D)** Western blots of glutamate receptor NMDA subunits 2A, 2B, and 1 (GluN2A, GluN2B, and GluN1), PSD-95, WAVE1, RhoA, and β-actin. α-tubulin is shown as control. Histograms to the right show the quantification as fold of untreated $IF1^{TG}$ mice ($n = 4$–6). **(E)** Total time spent exploring both objects in the training and test sessions of the long-term memory test ($n = 9$). **(F)** Representative track plots showing the position of the center of $IF1^{TG}$ mice during the open field test. Blue and red dots show the start and end points of the track, respectively. **(G)** No significant differences were found in distance traveled, number of lines of the virtual grid crossed, or time mobile ($n = 9$). Error bars: mean ± SEM. $^*P < 0.05$ by 2-tailed $t$ test (A, B, D, and E) or 2-way ANOVA with Bonferroni multiple comparisons test (C). In (C), data from $IF1^{TG}$ mice are replotted from S2A Fig, since the recordings were performed in the same experiment. Related to Fig 7. Uncropped western blots can be found in S1 Raw Images, and numerical data underlying plots in S1 Data. fEPSP, field excitatory postsynaptic potential; $IF1^{TG}$, IF1 overexpressing transgenic; MQ, MitoQ; PSD-95, postsynaptic density protein 95; RhoA, Ras homolog family member A; TBS, theta burst stimulation; WAVE1, WASP family member 1.
(TIF)

**S4 Fig. Overexpression of IF1 in neurons preserves exploratory activity, memory, and motor coordination in aged mice. (A)** Representative track plots showing the position of the center of approximately 2-year-old $IF1^{KO}$, CRL, and $IF1^{TG}$ mice in the open field test. Blue and red dots show the start and end points of the track, respectively. **(B)** Histograms show total distance traveled, number of lines of the virtual grid crossed, and time mobile. **(C)** Schematic of the long-term memory test. The plot to the right shows the discrimination index, calculated as the relative time spent exploring the novel object with respect to the total time exploring both objects. **(D)** No significant differences were found in total time exploring both objects. **(E)** The histogram shows latency to fall in the Rota-rod test. **(F)** Survival analysis by Kaplan–Meier and log-rank tests of $IF1^{KO}$ ($n = 6$), CRL ($n = 6$), and $IF1^{TG}$ mice ($n = 4$). The median life span of each genotype is shown to the right. Error bars: mean ± SEM for $IF1^{KO}$ ($n = 5$), CRL ($n = 4$), and $IF1^{TG}$ mice ($n = 5$). $^*P < 0.05$ by 2-tailed (B, E, and D) or pairwise (C) $t$ tests. Numerical data underlying plots can be found in S1 Data. CRL, control; IF1, ATPase inhibitory factor 1;

*IF1^{KO}*, IF1 knockout; *IF1^{TG}*, IF1 overexpressing transgenic.
(TIF)

**S1 Data. Numerical data.** Excel spreadsheet containing, in separate sheets, the underlying numerical data for Fig 1E–1J; Fig 2A–2J; Fig 3B and 3H; Fig 4C and 4D; Fig 5B and 5C; Fig 6C–6F, 6H, 6J and 6K; Fig 7A–7E; S1A–S1C Fig; S2A–S2C, S2E and S2F Fig; S3A–S3E and S3G Fig; S4B–S4F Fig.
(XLSX)

**S1 Raw Images. Uncropped western blots from all main and Supporting information figures.**
(PDF)

**S1 Table. Hippocampal metabolites.** The table summarizes the levels of other metabolites determined in hippocampal extracts. The results shown are the mean values ± SEM. Values are expressed as nmol per g, except for 3-hydroxyisobutyrate, 2-methyl-3-hydroxybutyrate, malate, and citrate, which are expressed as fold change of control. $^{*}P < 0.05$ with respect to control, $^{†}P < 0.05$ with respect to *IF1^{KO}* by 2-tailed *t* test. *IF1^{KO}*, IF1 knockout.
(DOCX)

## Acknowledgments

The authors thank Mrs. B. Sánchez-Garrido and M. Chamorro (CBMSO) for technical assistance. The authors thank the Confocal and Electron Microscopy, Proteomic and Genomics and NGS Facilities of CBMSO, and the CNB Genomics Facility for invaluable assistance.

## Author Contributions

**Conceptualization:** Pau B. Esparza-Moltó, Michael P. Murphy, José A. Esteban, José M. Cuezva.

**Data curation:** Pau B. Esparza-Moltó, Marta P. Pereira, Rafael Artuch.

**Formal analysis:** Pau B. Esparza-Moltó, José M. Cuezva.

**Funding acquisition:** José M. Cuezva.

**Investigation:** Pau B. Esparza-Moltó, Inés Romero-Carramiñana, Cristina Núñez de Arenas, Marta P. Pereira, Noelia Blanco, Beatriz Pardo, Georgina R. Bates, Carla Sánchez-Castillo, Rafael Artuch, Michael P. Murphy, José A. Esteban, José M. Cuezva.

**Methodology:** Pau B. Esparza-Moltó, Inés Romero-Carramiñana, Cristina Núñez de Arenas, Marta P. Pereira, Noelia Blanco, Beatriz Pardo, Georgina R. Bates, Carla Sánchez-Castillo, Rafael Artuch, Michael P. Murphy, José A. Esteban.

**Resources:** Cristina Núñez de Arenas, Rafael Artuch, Michael P. Murphy, José M. Cuezva.

**Supervision:** Rafael Artuch, Michael P. Murphy, José A. Esteban, José M. Cuezva.

**Visualization:** Marta P. Pereira.

**Writing – original draft:** Pau B. Esparza-Moltó, José M. Cuezva.

**Writing – review & editing:** Pau B. Esparza-Moltó, Inés Romero-Carramiñana, Marta P. Pereira, Beatriz Pardo, Rafael Artuch, Michael P. Murphy, José A. Esteban, José M. Cuezva.

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
