## [Editor Report · Decision Letter 0]

29 Jan 2021

Dear Dr Cuezva, 

Thank you for submitting your manuscript entitled "Mitochondrial reactive oxygen species generated by the IF1-inhibited pool of ATP synthase regulate cognition" for consideration as a Research Article by PLOS Biology.

Your manuscript has now been evaluated by the PLOS Biology editorial staff as well as by an academic editor with relevant expertise and I am writing to let you know that we would like to send your submission out for external peer review.

Please re-submit your manuscript within two working days, i.e. by Feb 02 2021 11:59PM.

Kind regards,

Lucas Smith, Ph.D.,

Associate Editor

PLOS Biology

---

## [Decision Letter · Decision Letter 1]

23 Mar 2021

Dear Dr Cuezva,

Thank you very much for submitting your manuscript "Mitochondrial reactive oxygen species generated by the IF1-inhibited pool of ATP synthase regulate cognition" for consideration as a Research Article at PLOS Biology. Your manuscript has been evaluated by the PLOS Biology editors, an Academic Editor with relevant expertise, and by several independent reviewers.

In light of the reviews (included below), we are pleased to offer you the opportunity to address the comments from the reviewers in a revised manuscript. As you will see, the reviewers are generally positive about your study, and call it ambitious, well planned, and well-executed. However, the reviewers have raised a number of points and concerns, including asking for more discussion of apparent discrepancies (reviewers 1 and 2), requesting additional mechanistic studies (reviewer 3) and additional control experiments (reviewers 1 and 2). We will leave it up to you how to best address these comments.

Reviewer 4 has asked for additional behavioral and electrophysiological analyses of the models used here, to assess how generalizable the effects are to synaptic function and cognition. Having discussed this comment with the Academic Editor, we think that while this analysis would be interesting, it may be beyond the scope of this current study. We would therefore not require these specific additional experiments in a revision, and think this point could be addressed via additional discussion.

**IMPORTANT In addition to addressing these reviewer comments, please also make sure to address the remaining editorial, data, and other policy-related requests included below my signature and summarized here:

1) It looks as though the financial disclosures statement got cut off and that there is a typo in the last word (manuscrip). 

2) Data Request: Please provide, as a supplementary file, the summary statistics used to generate each graph presented in your study. Please make sure to provide a legend for this file, and to refer to it in your data availability statement and figure legends (including supplementary figure legends). For example, to each figure legend, you could add a statement saying “The data underlying this figure can be found in the supplementary file S1_data”.

3) Data Request: Please provide as a supplemntary file all uncropped and minimally adjusted images for western blot and gel results reproted in the article. 

4) When resubmitting, please provide me with a way to access the proteomic and microarray data which has been deposited in online databases. 

We expect to receive your revised manuscript within 1 month, but please email us (plosbiology@plos.org) if you have any questions or concerns, or would like to request an extension. At this stage, your manuscript remains formally under active consideration at our journal; please notify us by email if you do not intend to submit a revision so that we may end consideration of the manuscript at PLOS Biology.

**IMPORTANT - SUBMITTING YOUR REVISION**

*Resubmission Checklist*

*Published Peer Review*

Sincerely,

Lucas Smith, Ph.D.,

Associate Editor,

lsmith@plos.org,

PLOS Biology

DATA POLICY:

Fig 1E-J; Fig 2A-J; Fig 3 B,H; Fig 4 C,D; Fig 5B-C; Fig 6 C-F,H,J-K; Fig 7A-E; Fig S1 A-C; Fig S2 A-C,E-F; Fig S3 A-E,G; Fig S4 B-F

Please upload the uncropped and minimally adjusted files for the following figures: Fig 1D,G,I; Fig 2G,J; Fig 6D,F; Fig 7B,D; Fig S1A-C; Fig S2B; Fig S3 A-B,D

REVIEWS:

Reviewer #1: In their manuscript titled "Mitochondrial reactive oxygen species generated by the IF1-inhibited pool of ATP synthase regulate cognition," Esparza-Moltó et al explore the implications varying doses of the mitochondrial protein IF1 on mitochondrial oxidative phosphorylation, neuronal function, and learning in mice. Previous studies with full-body IF1 KO mice had yielded conflicting results regarding its function and the effect of its KO. In this study, the authors generated forebrain-specific IF1 KO and human-IF1-overexpressing mice. As expected, they observed decreased ATP synthase activity, increased membrane potential, decreased oxygen consumption, and increased ROS with increasing IF1 dose. Through transcriptomics and proteomics, the authors identified changes in the synaptic components as substantially affected by IF1 dose. The authors then examine postsynaptic response in the glutamatergic synapse between CA3 and CA1 neurons finding it increased with increased levels of IF1 and with a corresponding effect on memory. Finally, by using the antioxidant MitoQ the authors found that some of the memory benefits gained from higher levels of IF1 may be the result of IF1-driven ROS generation and ROS signaling. This is well-planned, well-written study that elegantly links subtle changes in respiration to synaptic function and behavior. 

Minor points:

1- Human IF1 is used. The authors should comment on the conservation of IF1 between mouse and human and comment on any effect that human rather than mouse IF1 for overexpression may have. 

2- On page 5, the authors note that cristae are disorganized in the IF1 KO mice. They also note later that the ATP synthase oligomeric state is altered in the IF1 KO. Given, ATP synthase dimers affect cristae structure do the authors believe this to be a potential mechanism of the cristae abnormalities. The authors should comment on why cristae are observed to be disorganized in the IF1 KO.

3- The authors establish that IF1 is knocked out in the forebrain, but as this is a forebrain-specific conditional knockout, it needs to be shown that IF1 levels are unaffected in other tissues such as cerebellum or liver. 

4- There is not a perfect correlation between the electrophysiology in 6A and the behavior in 6G- K, in that the KO has similar electrophysiology to the WT but a memory deficit at 2 hrs compared to WT. The authors should explicitly address reasons for this discrepancy in the discussion. 

5- Fig. S2B shows decreased levels of NMDA receptor subunits in IF1 TG mouse samples. Does the gene expression data show decreased expression? If not, might this be mediated by receptor endocytosis and degradation due to increased stimulation by glutamate in IF1 TG mice (Fig. 5 shows increased glutamate levels in IF1 TG)? It would be helpful to state why NMDA receptor subunit levels are reduced in IF1 overexpression.

Reviewer #2: In this article by Esparza-Moltó et al., the authors have investigated the effects of overexpressing and knocking out the protein IF1, an inhibitor of the ATP synthase. They have taken multi-pronged approaches to characterize the impact of IF1 knockout and overexpression on mitochondrial structure, function, neuronal transcriptomics, proteomics, metabolomics, electrophysiology, and cognitive tasks. I recommend this paper for publishing provided the following questions are addressed:

1. Figure 1H: at higher IF1 levels, the mitochondria structures are elongated, and in the knockout condition, they are rounder. Is that fragmentation? If yes, isn't mitochondria fragmentation an indicator of glycolytic enhancement for ATP synthesis? Could you please elaborate?

2. Figure 1E: the experiment should be performed in the presence of a glycolysis inhibitor + mitochondrial substrates (pyruvate or lactate) to show that the observed ATP synthesis increase is due to an increase in mitochondrial ATP production and not glycolysis.

3. Could you please explain how IF1 increase reduces glycolytic flux? For instance, are glucose transporters reduced on the membrane surface that results in lower glucose intake?

4. Figures 3 and 4: the proteomics and transcriptomics analyses results denote downregulation of LTP in IF1 overexpressed neurons. However, the electrophysiology measurements and the animal behavior experiments show otherwise. Please explain.

5. The resolution of the figures needs to be improved. Some of the font sizes also need to be made more prominent.

Reviewer #3: In this overall well-executed study, the authors used newly generated genetic strains of ATPase inhibitory factor 1 (IF1) to characterize the effects of loss- and gain-of-function of IF1 in the mouse. Results clearly showed that IF1 interacts with ATPase under physiological condition to define the fraction of active/inactive enzyme in vivo, and that IF1 dose critically regulates mitochondrial metabolism, synaptic function, and learning and memory. Moreover, the main effect of IF1 appears to be mediated by mitochondrial ROS, as the treatment of IF1 Tg mice with the mitochondrial-targeted antioxidant mitoQ effectively blocked the synaptic and cognitive effects of IF1 overexpression in the Tg animals. Overall, the experiments are well executed with sufficient statistical analysis, the conclusion is well supported, and the take home message is clear: Regulation of mitochondrial ROS signaling by IF1 is important for synaptic function and cognitive behavior in mammals.

One point that is missing from the study is the mechanism of mitochondrial ROS regulation by IF1. Since mitochondrial complex I is a major source of mito-ROS, and one way for IF1 to affect ROS production by complex I is through reverse electron transfer (RET). It would add more mechanistic insight to the study if the authors can test the involvement of RET in IF1-induced mito-ROS production using RET inhibitors described in the literature.

Reviewer #4: Review of Esparza-Moltó

This is an ambitious paper that makes the case that ATPase inhibitory factor 1 (IF1), an inhibitor of mitochondrial ATP synthetase, regulates Esparza-Moltó synaptic function, learning and memory. Such a claim deserves close consideration and has bold implications because it amounts to saying that regulating ATP availability, a fundamental cellular metabolic function is sufficient to control neural circuit and network function. There are so many steps along the causal chain between ATP synthetase regulation and expression of memory, that it is head spinning to even contemplate, but the authors do a remarkable job of making a plausible case. The experiments are laid out like detectives aiming to meet the burden of proof of "a preponderance of the evidence" rather than a "beyond a reasonable doubt." In short, the work is compelling, but it leaves this reader with the uncomfortable conclusion that it is hard to understand how regulating ATP availability, tantamount to energy availability, leads to the cellular and behavioral manifestation of knowledge. Nonetheless, the evidence of a series of carefully selected and logical correlations is remarkable, and while the biochemistry and respiratory signaling is not my area of expertise, the data and implications are intriguing even if implausible.

I was especially impressed that the commercial antioxidant MitoQ, which like endogenous antioxidant CoQ10, reduces mitochondrial reactive oxygen species was able to block the effects of overexpression of IF1 in transgenic mice, arguing that quenching ROS, a result of ATP metabolism, is sufficient for regulating cellular functions like synaptic strength at the hippocampus Schaffer collaterals, and object recognition memory. 

While I confess, that I do not see any outright logical flaws, the conclusions are still unsettling and their generality questionable. Consequently, I'd like to see the same experiments done for another synapse and another behavior, perhaps a conditioned memory to understand how specific or general these effects are. Given that energy metabolism is so basic, one imagines the effects must generalize to many synapses and behaviors. I should point out that regardless of the result, the work deserves to be published, but it would be misleading as is, to publish work that implies such general effects without demonstrating the generality.

The Schaffer collateral fEPSP waveforms are unusual. If the scale bars of 10 ms in Figs. 6 and 7 are correct then the responses are very slow. What is the Fig. 6B scale bar? It shouldn't take 10 ms for the fiber volley! Similarly, the fEPSP slopes seem to be off by an order of magnitude. More importantly, the waveshape of the fEPSP response are unusually complex in the transgenic and knockout mice. The "ringing" of population spikes suggest the slices are epileptic. That would make it very difficult to accurately measure the synaptic response, given ongoing seizure-like activity. The picrotoxin to block inhibition, likely has exacerbated this, but was unnecessary. In short, but consistent with the fundamental disruption that one expects by manipulating IF1, these mutant mice (or at least their hippocampal slices) do not appear to be normal in more fundamental features than the features that are measured by the stimulus response assessment.

---

## [Editor Report · Decision Letter 2]

29 Apr 2021

Dear Dr Cuezva,

Thank you for submitting your revised Research Article entitled "Mitochondrial reactive oxygen species generation is enhanced by the IF1-inhibited pool of ATP synthase and regulates cognition" for publication in PLOS Biology. On behalf of my colleagues and the Academic Editor, Franck Polleux, I am pleased to say that we can in principle offer to publish your Research Article "Mitochondrial reactive oxygen species generation is enhanced by the IF1-inhibited pool of ATP synthase and regulates cognition" in PLOS Biology. We have discussed your revised manuscript with the Academic Editor, and think that it is has satisfactorily addressed the reviewer concerns. We think that the study is well executed and that the results will be of broad interest. 

Before we can publish your paper, we will need you to address any remaining formatting and reporting issues. These will be detailed in an email that will follow this letter and that you will usually receive within 2-3 business days, during which time no action is required from you. Please note that we will not be able to formally accept your manuscript and schedule it for publication until you have made the required changes.

We would also like to suggest that you streamline the title of your manuscript a bit. For example, we think it could be shortened to something like "generation of mitochondrial reactive oxygen species is controlled by ATPase inhibitory factor 1 and regulates cognition". If you agree, during the last formatting checks, please provide an updated manuscript with an edited title. 

PRESS

Thank you again for supporting Open Access publishing. We look forward to publishing your paper in PLOS Biology. 

Sincerely, 

Lucas Smith, Ph.D. 

Senior Editor 

PLOS Biology